# *Drosophila* ovarian stem cell niche ageing involves coordinated changes in transcription and alternative splicing

Dilamm Even-Ros [1], Judit Huertas-Romero [1], Miriam Marín-Menguiano [1], Gretel Nusspaumer[1], Miguel Borge [2,3], Manuel Irimia [2,3,4], Federico Zurita [5] ✉ & Acaimo González-Reyes [1] ✉

Gene expression (GE) and alternative splicing (AS) contribute to the formation of new interaction networks with potentially significant cellular functions. Here, we investigate ageing in the *Drosophila* female germline stem cell (GSC) niche and describe functional changes in both GE and AS. The GSC niche comprises three types of support cells, whose ageing transcriptomes reveal differential GE and AS variations related to cell adhesion, cytoskeleton and neural signalling. Because each population show distinctive GE and AS changes, niche cell types possess unique ageing signatures. Depending on the cell population, groups of genes display changes in both GE and AS, revealing a coordinated regulation of transcription and splicing during niche ageing. One such gene is *Fasciclin 2*, a neural adhesion molecule that we find is essential for niche functioning. Furthermore, genes involved in AS undergo changes in GE and/or AS themselves, providing a mechanistic explanation for the coordination of these two processes during niche ageing. This is the case of the splicing factor Smu1, described here as a key element necessary for ovarian niche homeostasis.

The healthy function of adult organs in animals requires controlled homeostasis during tissue maintenance or in response to pathophysiological situations. As a general principle, tissue function relies on populations of adult stem cells capable of making most of the cell types found therein. Present in most tissues studied to date, stem cells often reside in specialized microenvironments (niches) known to regulate their proliferation and differentiation. During ageing, the extent to which tissues maintain proper homeostasis and regenerative capacity depends on the activity of resident stem cell populations. In fact, impairment of stem cell-niche activity and/or numbers results in tissue attrition, a hallmark of ageing in which tissue function declines as a consequence of unbalanced replenishment of cells lost to wear or injury[1,2]. Ageing can provoke accumulation of somatic mutations and

clonal expansion of stem cells, as shown for the hematopoietic system. It can also induce a decrease in their proliferative capacity and stem cell loss[3–5]. At the cellular level, ageing is accompanied in stem cells by an increase in genotoxic stress (including higher concentrations of reactive oxygen species) and by changes in epigenetic regulation and genomic integrity, cell adhesion, responsiveness to signalling, cell polarity, nutrient sensing and self-renewing divisions[6–13].

In spite of its importance, the mechanisms by which the regenerative potential of tissues is diminished during ageing are still unclear, particularly so in those stages in which the organism is within reproductive age. In this context, the case of germline stem cell (GSC) ageing is critical, as these cells give rise to entire new organisms probably after rejuvenating with each generation[14]. Thus, the quality of GSCs should

[1]Centro Andaluz de Biología del Desarrollo (CABD), CSIC-Junta de Andalucía-UPO, Carretera de Utrera km 1, 41013 Seville, Spain. [2]Centre for Genomic Regulation, Barcelona Institute of Science and Technology (BIST), Barcelona, Spain. [3]Universitat Pompeu Fabra (UPF), Barcelona, Spain. [4]ICREA, Barcelona, Spain. [5]Departamento de Genética e Instituto de Biotecnología, Universidad de Granada, Centro de Investigación Biomédica, 18071 Granada, Spain. ✉e-mail: f.zurita@ugr.es; agonrey@upo.es

not be compromised during reproductive age. Still, there are examples of changes in the behaviour of ageing stem cell populations in fertile adults such as the *Drosophila melanogaster* gonads, and mammalian spermatogonial stem cells can function much longer than a lifetime when transplanted into younger recipients[8,15–18]. This evidence led to the proposal that germline niche ageing appears to affect mainly niche support cells rather than stem cells themselves[1].

Alternative splicing (AS) is a posttranscriptional mechanism that increases proteome complexity—and hence an expansion of protein interaction capabilities—by variable precursor mRNA processing from a given set of transcribed genes. This general characteristic of eukaryotic genomes, in which 90–95% of genes could show alternative splicing, as in the case of humans, includes the generation of mRNAs with alternative splicing sites, exon skipping, mutually exclusive exons and retained introns[19]. During development, AS networks contribute to the acquisition of organ identity, as shown for heart, brain or liver[20–23], even though the correspondence between transcriptomes and proteomes could show poor correlation[24,25]; but see refs. 26,27. Moreover, splicing changes with age in a number of animal models, thus potentially linking compromised physiological homeostasis with the quality of transcriptomes[28]. However, whether splicing affects stem cell niche ageing is still unknown. In an effort to identify novel mechanisms involved in stem cell niche ageing and to correlate differential GE and AS with stem cell niche homeostasis, in this work we make use of a well-studied cellular niche amenable to cellular and genetic manipulation, that of the *Drosophila* ovary. We have focused our efforts on the changes occurring in GE in ovarian niche support cells during the course of ageing, with an emphasis on the role of AS in the process. In addition to alternative splicing, our work also identifies cell polarity, cell adhesion and mechanisms of neural signalling and development as targets of ageing in GSC niche cells.

## Results

The germline niche of the female fruit fly is a simple arrangement in which extracellular matrix and three somatic cell types—terminal filament cells (TFCs), cap cells (CpCs) and anterior escort cells (ECs)—sustain a population of 2–4 GSCs. The GSC niche is found in the germarium, a tapered structure at the anterior tip of the ovarioles, structural units that conform the adult ovaries. Posterior to the GSC niche reside the follicle stem cells (FSCs), responsible for the generation of follicle cell precursors[29–31] (Fig. 1a). The combined action of both stem cell populations and their accompanying cells is behind the constant production of new egg chambers throughout the first weeks of the adult female. We have employed bulk RNA sequencing (RNA-seq) to identify candidate genes involved in niche cell ageing. To obtain a comprehensive description of niche ageing, we made use of genetic tools that allowed us to profile and analyse separately two groups of niche support cells, TFCs-CpCs and ECs. We focused our efforts on both GE variations and AS events that take place in the first four weeks of the female life. We chose this time frame because the number of GSCs hosted within the ovarian niche was significantly reduced at 3- and 4-weeks after eclosion ($P < 0.0001$; Fig. 1b). However, at least as indicated by several studies[32–35], 4-week old females do not show a decrease in survival rates and are fully fertile, suggesting that the overall fitness of adult females this old is not compromised and our chosen time-frame avoids indirect, systemic ageing effects on ovary homeostasis.

### Ageing TFCs-CpCs show changes in GE and AS profiles
We first isolated TFCs and CpCs from 1 week- (1w) and 4w-old females, performed RNA-seq and analysed GE profiles (Fig. 1c). We could identify 1421 genes whose expression was either up- (649) or down-regulated (772; Fig. 1d; Supplementary Data 1) in 4w-old cells when compared to younger 1w-old samples (fold change <0.5 or >2; false discovery rate <10%). Gene Ontology (GO) enrichment analyses of the identified genes yielded a number of categories with enriched genes

associated to cell adhesion, cell migration, cell polarity, cytoskeleton, extracellular matrix, ion transport, muscle or neural signalling/development (Fig. 1e; Supplementary Fig. 1a). While it was surprising to find expression of genes characteristic of cell types or processes not represented in our samples (i.e., muscle, neurons or migrating cells), some of the selected genes have known functions in a myriad of cell types, including ovarian lineages. For instance, *Fasciclin 2* (*Fas2*) was identified as a gene whose expression was up-regulated in 4w-old cells. Fas2, the *Drosophila* orthologue of neural cell adhesion molecule (NCAM), is required for proper neural synapsis and is known to mediate activity-mediated synaptic plasticity, stabilisation and growth. It is also expressed during oogenesis[36–41]. In addition, we have validated the expression in TFCs-CpCs of genes differentially expressed during ageing in our RNA-seq but not known previously to be present in the germarium, such as MSP300 or *muscleblind* (Supplementary Fig. 1b, c). Finally, the identification of genes encoding ion channels whose expression varied with age such as *Pkd2 (Polycystic kidney disease 2)* and *Clic (Chloride intracellular channel)*, together with genes involved in neural signalling and development or cytoskeleton organisation such as *Nlg1 (Neuroligin 1)* or *Actn (α actinin*; Supplementary Data 1), may indicate that TFCs-CpCs could be excitable and/or respond to ion fluxes, much as neurons or muscle cells do[42,43].

Next, we studied AS events using vast-tools[22,44]. We quantified AS changes in five different categories, namely exon skipping and mutually exclusive exons (clustered under the "alternative exons" group), alternative splice acceptor and donor site choices, and retained introns (Fig. 2a). The comparison of 1w- *versus* 4w-old TFCs-CpCs identified 958 events corresponding to 673 genes, with an absolute differential percent splicing inclusion (dPSI)>15 (Supplementary Data 1). Young and aged samples showed a significantly different distribution of events involving exon usage ($P < 0.008$) and the retention of introns ($P < 0.001$; Fig. 2b). Thus, older samples exhibited a tendency to reduce the inclusion of alternative exons and to increase intron retention. Considering the multi-level regulation by intrinsic and extrinsic cellular factors of exon usage and intron retention[19], our finding likely reflects a specific programme for mRNA splicing during TFCs-CpCs ageing.

The examination of GE and AS events yielded a number of interesting observations. First, 153 (22.7%) genes displaying differential GE also showed AS modifications during ageing (254 events, with a distribution of AS categories similar to the complete dataset; Chi-square test $P > 0.05$; Fig. 2b, c). This is in contrast to most comparative analyses of GE and AS, which have shown that the two layers mainly regulate non-overlapping genes (e.g. ref. 45). Second, in line with this overlap, the analysis of cellular components and biological processes GO terms corroborated that changes in GE and in AS affected common, essential categories. Thus, the cell adhesion, cell polarity, cytoskeleton, ion transport, muscle and neural signalling/development categories were also represented in the AS dataset (Figs. 1e, 2c, d; Supplementary Fig. 2a; Supplementary Data 1). Third, a KEGG pathway analysis of the genes affected by changes in AS in ageing cells identified nine cellular or signalling pathways, all of which are associated with ageing. Remarkably, we found that several genes associated to six of the above pathways were also differentially expressed during ageing in TFCs-CpCs (Fig. 2e; Supplementary Fig. 2b). Fourth, a Cumulative Distribution Function (CDF) analysis determined that the genes with differential AS tend to be more downregulated than the rest (Supplementary Fig. 2C, D; $P < 0.0001$). In all, our results demonstrate that ageing of the TFC and CpC populations in the ovarian stem cell niche comprises not only significant variations in GE but also in AS. Because an overlapping collection of genes presented changes in both, we surmise that the ageing of TFCs and CpCs requires the coordination of transcriptional changes with specific differential splicing events that involve exon usage, intron retention events and the use of alternative acceptor and donor splice sites.

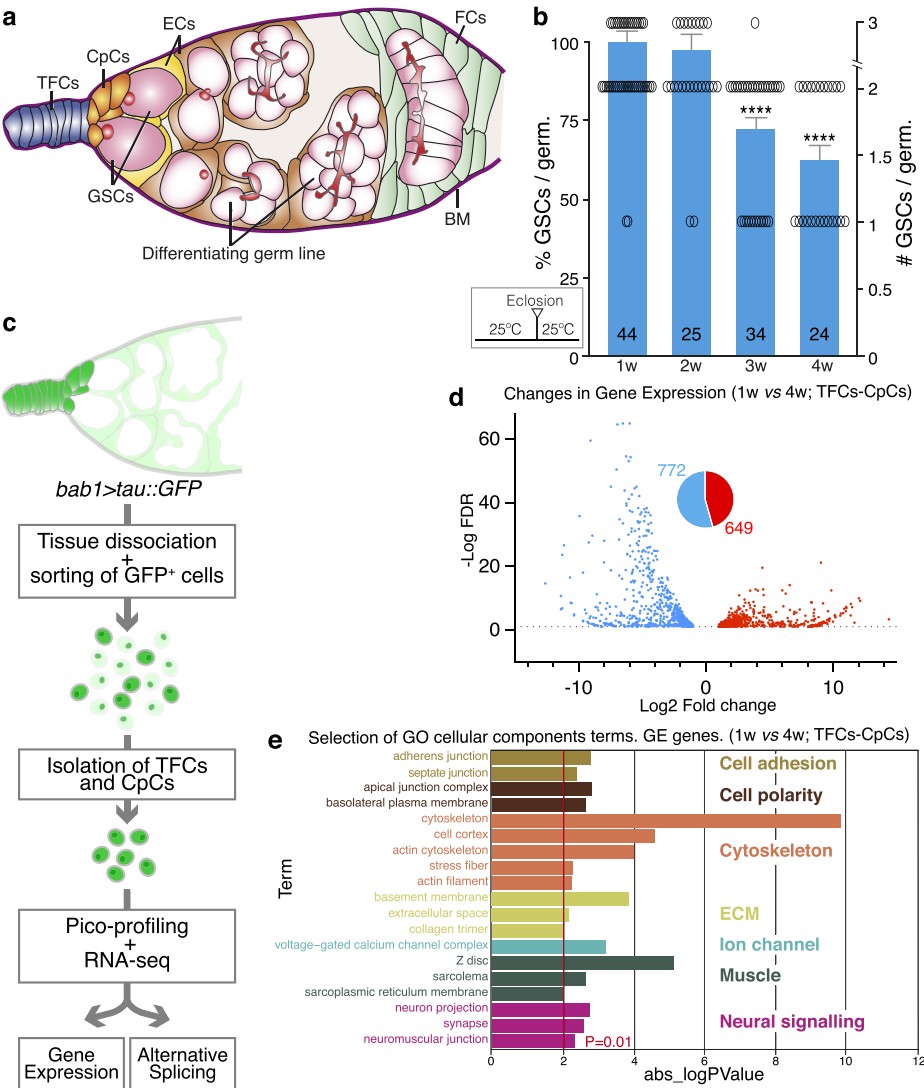

**Fig. 1 | Gene expression (GE) analysis during TFC and CpC ageing. a** Scheme of the GSC niche showing the surrounding basement membrane (BM) and the disposition of the terminal filament cells (TFCs), cap cells (CpCs), GSC-associated escort cells (ECs; yellow) and more posterior ECs (brown), follicle cells (FCs), germline stem cells (GSCs) and differentiating germline cysts. GSCs display an anterior, round spectrosome and the differentiating cysts contain branched fusomes (pink). **b** Graph depicting the number of GSCs per niche in 1-, 2-, 3- and 4-week old control germaria raised at 25 °C. Numbers in the columns denote the sample size for each of the age groups (*n*). Statistically significant *P* values of two-tailed, unpaired *t*-tests of different age groups compared to 1-week old females are shown (**** 3w: *P* = 6.63E−07, **** 4w: *P* = 1.17E−08). **c** Workflow undertaken to isolate TFCs-CpCs for GE and alternative splicing analyses. Germaria expressing Tau::GFP under the control of the *bab1-Gal4* line in TFCs, CpCs and ECs (albeit at

lower levels in the latter) were used to sort cells expressing high GFP levels (TFCs and CpCs) followed by pico-profiling and RNA-seq. **d** Changes in GE−represented as log2 fold change at 4 weeks/1 week *versus* the -log10 false discovery rate (FDR)− identified 649 genes upregulated (red) and 772 (blue) genes downregulated in 4-week old compared to 1-week old cells (linear fold change <0.5 or >2; FDR < 10%). Data correspond to two biological replicates per time point. **e** Representation of a selection of terms identified in the GO-cellular components analysis. ECM: extracellular matrix. In this and the rest of the GO terms analyses, *P* values were calculated using the EASE modification of Fisher's Exact test (https://david.ncifcrf.gov) with a cut-off value of *P* ≤ 0.01 (absolute log10 *P* value ≥2). Error bars in **B** indicate the standard error of the mean (SEM). Related to Supplementary Fig. 1, Supplementary Data 1 and Supplementary Data 4.

## EC ageing involves specific changes in GE and AS profiles

To determine whether the molecular events that characterised the ageing of TFCs-CPCs were general to all ovarian niche cells, we analysed in detail the transcriptome of ECs of different ages. We isolated 1w- and 4w-old ECs (Supplementary Fig. 3a) and confirmed by RNA-seq significant alterations in the patterns of gene expression and in their splicing programme with age. There were however striking differences with the TFCs-CpCs populations. First, the number of genes showing significant changes in GE was markedly smaller in ECs (72 down-regulated at 4w compared to 1w-old cells; 50 genes up-regulated; fold change <0.5 or >2; false discovery rate <10%). Second, we identified 769 AS events (corresponding to 639 genes; dPSI >15),

which identified few GO cellular components terms associated to cell adhesion, cytoskeleton or nucleus. While these numbers were comparable to those found in the TFCs-CpCs dataset, only 5 (0.8%) genes with variations in AS displayed differences also in GE. Third, it was young EC samples that were significantly biased towards intron retention events, in contrast to the TFCs-CpCs type, in which it was the old cells the ones that showed higher incidence of intron retention events. Since intron retention is a significant, controlled event in the regulation of gene expression and therefore in cell physiology[19], the fact that ECs exhibit opposite patterns of IR events to TFCs-CpCs (older ECs have significantly fewer IR events than younger ones) strongly suggests cell type-specific ageing programmes in place

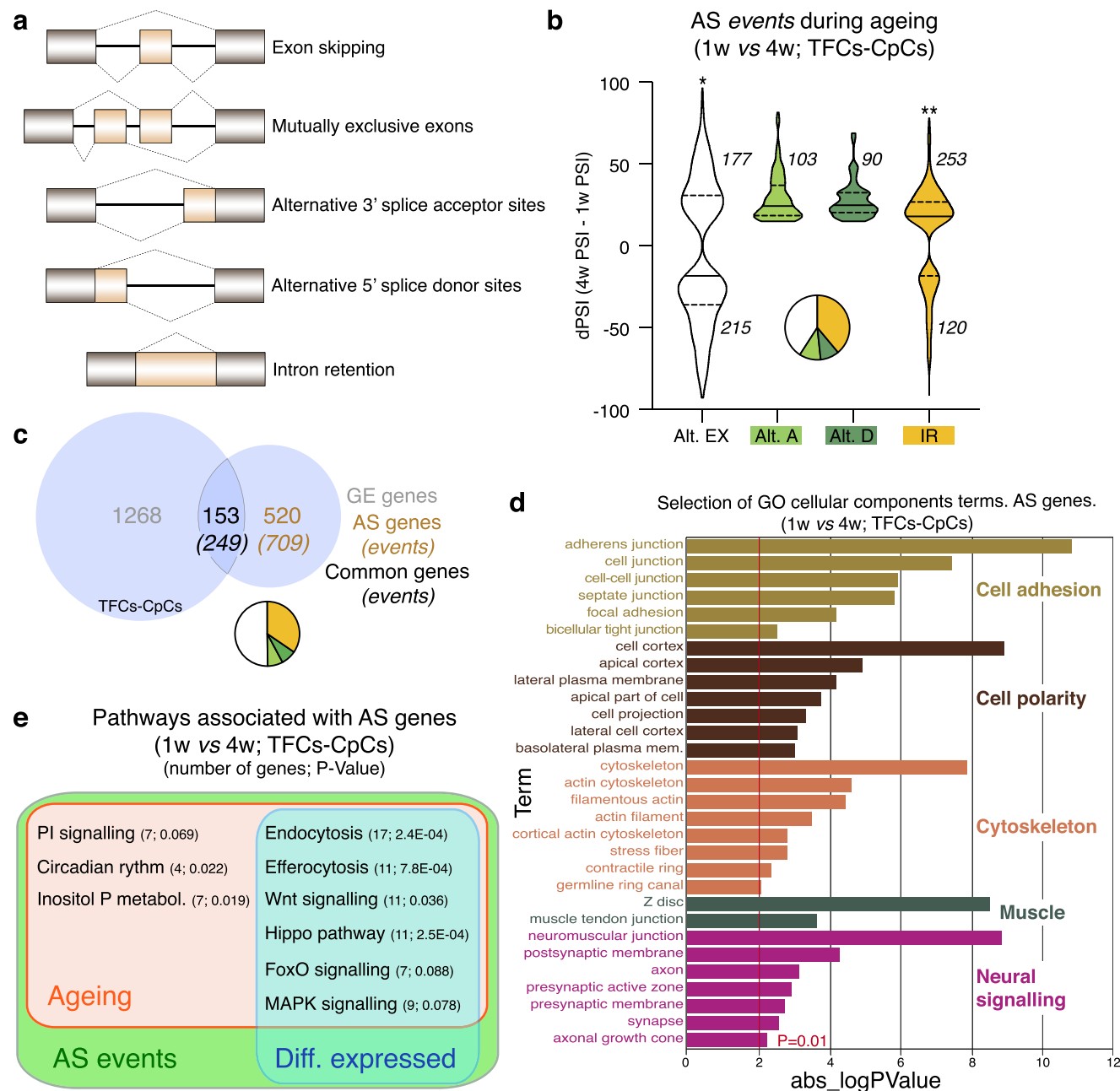

**Fig. 2 | Alternative Splicing (AS) events during TFC-CpC ageing. a** Scheme showing the different AS categories analysed in this study. **b** Graph showing the distribution of the 958 changes (affecting 673 genes) in AS events during TFC-CpC ageing identified in our studies (in this and the rest of figures, Alt. E: alternative exons refer to skipped and/or mutually exclusive exons; Alt. A or D: alternative acceptor or donor splice sites correspond to variations in 3′ and/or 5′ splice-site selections; IR: intron retentions; medians are shown as horizontal lines in the boxes; dotted lines represent first and third quartiles). Changes are represented as variations in percent splicing inclusion (dPSI=4w PSI − 1w PSI). Only events with an absolute dPSI>15 were considered. Data correspond to two biological replicates per time point. Statistically significant *P* values of the compare proportions test are shown for the Alt. EX and IR categories (*P = 0.008; **P = 0.001). **c** Diagram to indicate the number of genes showing changes in GE and in AS during TFCs-CpCs

ageing. Pie charts represent the different AS categories. **d** Representation of a selection of terms identified in the GO-cellular components analysis. Of note, the "cell adhesion", "cell polarity", "cytoskeleton", "muscle" and "neural signalling" categories are shared with the cellular components groups identified in the GE analysis. **e** Among the genes with changes in AS, we could identify components of nine pathways with a significant enrichment associated to the corresponding KEGG gene lists, all of which were related to ageing. Six of them also contained genes that were differentially expressed in aged TFCs-CpCs, as determined in our GE analysis. In this and the rest of the KEGG pathway analyses, *P* values were calculated using the Fisher's Exact test. Pie charts indicate the proportion of the different AS categories in all the genes showing AS changes (**b**) or the genes displaying both GE and AS variations (**c**). Related to Supplementary Fig. 2 and Supplementary Data 1.

(Fig. 3a–c, e). Altogether, these results indicate that EC ageing involves fewer changes in GE than TFCs-CpCs and that both GE and AS in ECs seem to affect different gene pools. This is confirmed by the fact that all the cellular or signalling pathways associated with AS events (KEGG analysis) did not include genes with GE variations and

that the CDF analysis did not give significant differences between controls and genes with differential AS. In fact, the only biological processes terms common to both GE and AS groups corresponded with muscle organization/development (Fig. 3d, Supplementary Fig. 3b–f; Supplementary Data 2).

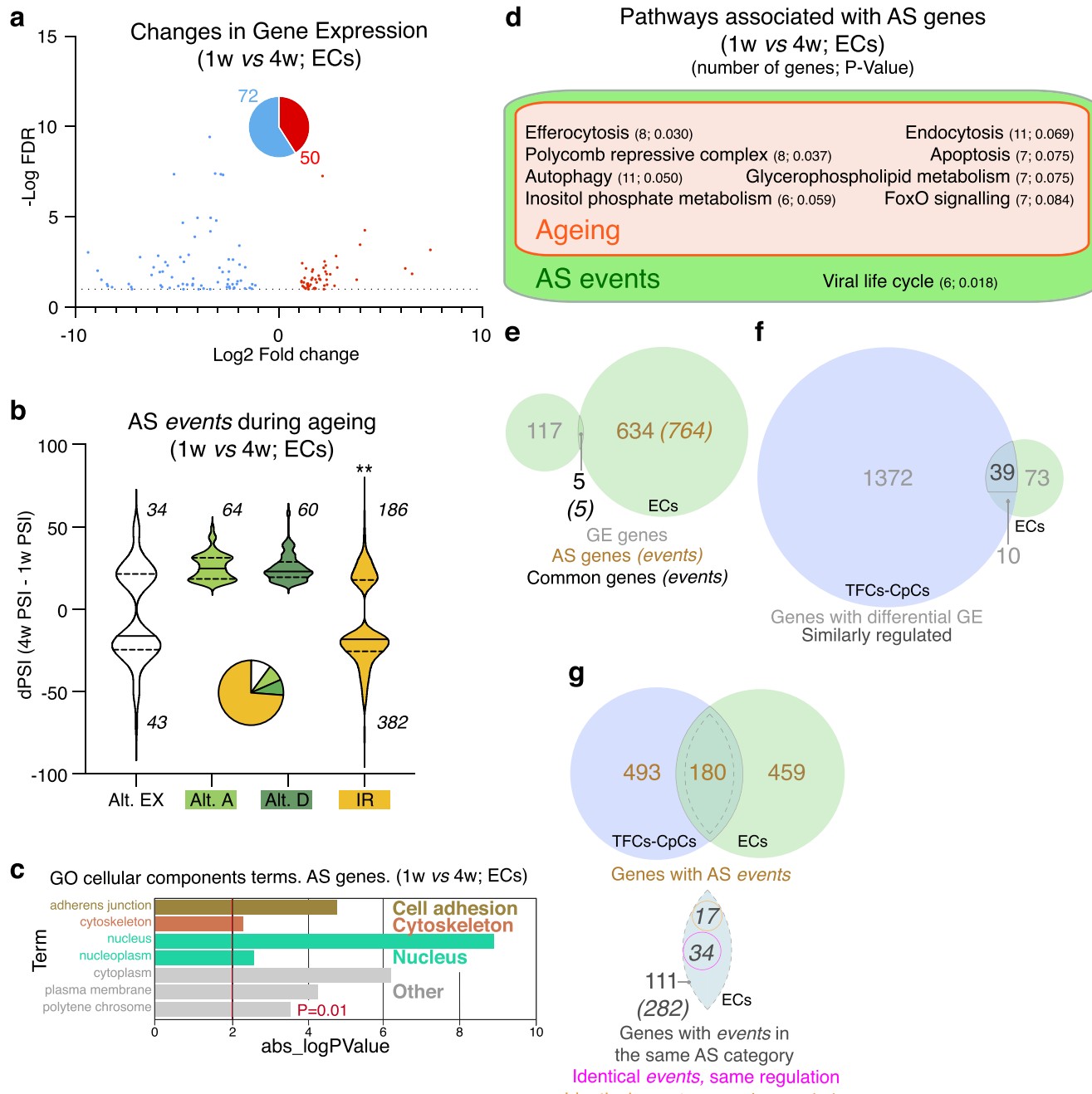

**Fig. 3 | Differential GE and AS events during EC ageing. a** Changes in GE—represented as log2 fold change 4 weeks/1 week *versus* the -log10 false discovery rate (FDR)—identified 50 genes upregulated (red) and 72 (blue) genes downregulated in 4-week old compared to 1-week old cells (linear fold change <0.5 or >2; FDR < 10%). **b** Distribution of the 769 changes (affecting 639 genes) in AS events during EC ageing. Changes are represented as variations in percent splicing inclusion (dPSI=4w PSI – 1w PSI). Only events with an absolute dPSI>15 were considered. Statistically significant *P* value of the compare proportions test is shown for the IR category (**$P < 0.001$). Pie chart represents the different AS categories. **c** Representation of a selection of the terms identified after a GO-cellular components terms analysis of AS genes. We could not identify any significant term after a similar analysis with the GE gene list (see Supplementary Data 2). **d** Among the genes with changes in AS, we could identify components of nine cellular or signalling pathways, eight of them related to ageing. None of the nine pathways contained genes that were differentially expressed in aged ECs, as determined in our GE analysis. **e** Diagram to indicate the number of genes showing changes in both GE and AS during EC ageing. The five overlapping genes are *CG44085*, *GstS1*, *MMP1*, *Pex1* and *Psa*. **f** Diagram representing the number of differentially expressed genes in the TFCs-CpCs and ECs datasets. Of the 49 genes common to all cell types, 39 exhibited the same GE variation (either up- or down-regulated). **g** Diagram representing the number of genes showing AS changes in TFCs-CpCs and ECs datasets. Of the 180 common genes, 111—corresponding to 282 events (in italics): share events in the same AS category. Of the 282 events, 51 are identical in both groups, 34 of them with the same change (either up- or down-represented; alternative 5′ and/or 3′ splice-site selections are included in this subgroup). Data correspond to two biological replicates per time point. Related to Supplementary Fig. 3, Supplementary Data 2 and Supplementary Data 4.

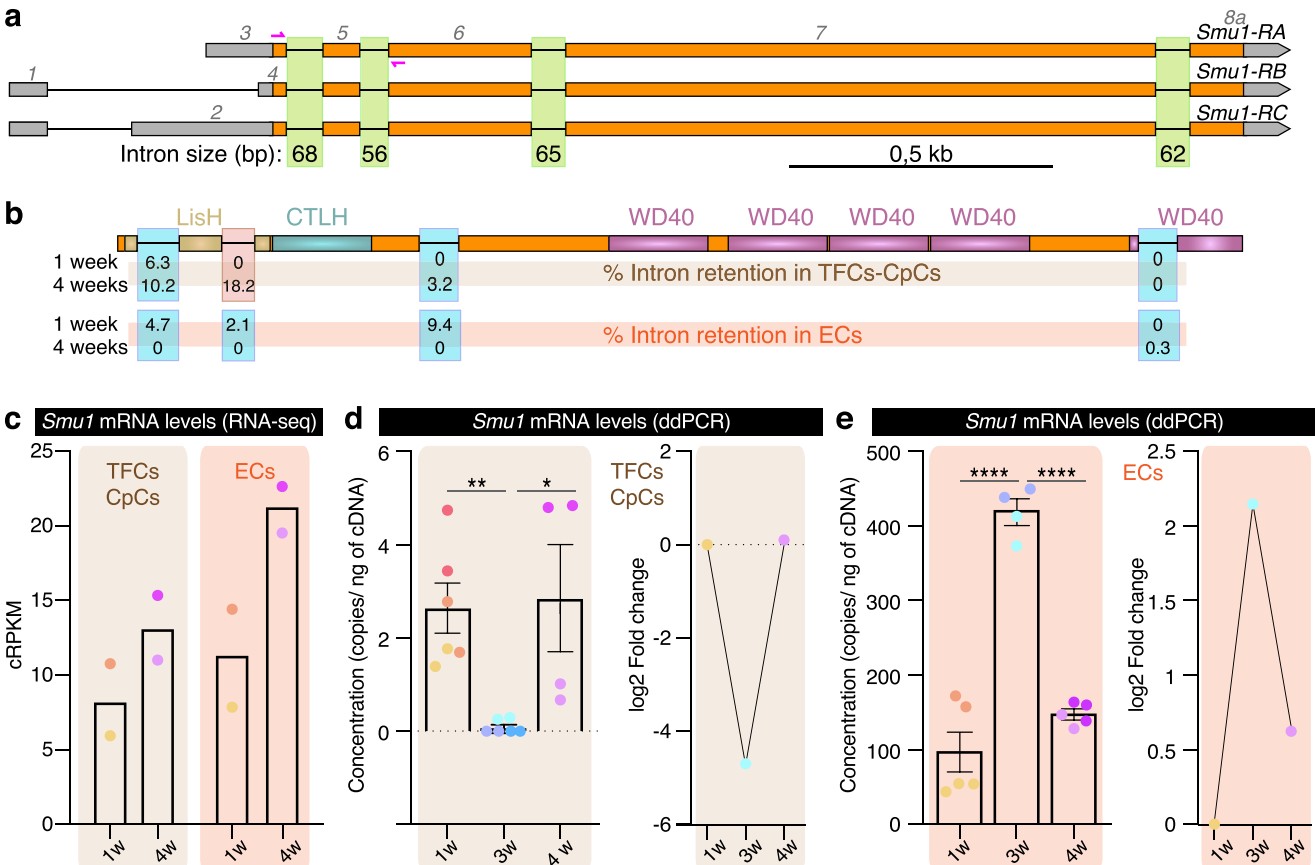

**Fig. 4 | Smu1 expression during ageing in niche cells. a** Genomic organization of the *Smu1* gene. While the three isoforms share the same coding region, the 5'UTRs vary. **b** Protein domains found in Smu1 (LisH: LIS1 homology; CTLH: C-terminal to LisH; WD40: tryptophan-aspartic acid 40). In 4-week old TFCs and CpCs, the second intron is retained in 18.2% of the transcripts, compared to 0% in 1-week old cells. In the case of ECs, none of the four introns are retained above the dPSI>15% threshold. **c** Quantification of *Smu1* mRNA levels with RNA-seq in TFCs-CpCs and ECs, represented as cRPKM (corrected reads per kilobase of target transcript sequence per million of total reads;[77]) Quantification of *Smu1* mRNA levels with droplet digital PCR (ddPCR) in TFCs-CpCs (**d**) and ECs (**e**), represented as number of transcript copies per nanogram of cDNA and the log2 of the fold change in expression levels at 3- and 4-weeks with respect to 1-week levels. Data in (**b**, **c**)

correspond to two biological replicates per time point. Following editorial guidelines, statistics such as error bars or *P* values cannot be derived from fewer than three values. Data in (**d**) correspond to two technical replicates of three (1- and 3-weeks) or two (4-weeks) biological replicates. Data in (**e**) correspond to two (3-weeks) or three (1- and 4-weeks) technical replicates of two biological replicates per time point. Same colour dots represent values of the same biological replicates. Error bars indicate SEM. *P* values of two-tailed, unpaired *t*-tests considered statistically significant between different samples are shown (* 3w *versus* 4w TFCs-CpCs: *P* = 0.0166, ** 1w *versus* 3w TFCs-CpCs: *P* = 0.0007, **** 1w *versus* 3w ECS: *P* = 3.83E−05, **** ECS 3w *versus* 4w: *P* = 8.07E−07). Related to Supplementary Fig. 4, Supplementary Data 1 and Supplementary Data 2.

Next, we sought to define the set of genes that presented similar behaviours during ageing in the different niche cell types. Of the 122 genes showing GE changes in ECs, 49 (41.6%) were also changed in TFCs-CpCs and, of these, 39 had the same expression profile (either up- or down-regulated). In the case of genes showing AS changes, of the 673 identified in TFCs-CpCs and the 639 in ECs, only 180 genes were common. Of these, 111 (corresponding to 282 events) showed AS variations in the same categories (exon usage, alternative splice acceptors and donors, and/or intron retention). Of the 282 AS events, 51 were identical, 34 of which were similarly regulated, and 17 had the opposite regulation (Fig. 3f, g; Supplementary Data 2). In all, we conclude that TFCs-CpCs and ECs present different ageing profiles in which a few, common genes change in both cell groups but with the vast majority of AS events being cell-type specific. Surprisingly, in spite of undergoing limited cell migration, cell replacement and of possessing a more complex morphology and genetic diversity than TFCs-CpCs (there are at least three subtypes of ECs defined by transcriptomic analyses)[31,46,47], EC populations have a more stable transcriptome in terms of GE changes during ageing than TFCs-CpCs, at least in the first four weeks of adult life.

## Smu1 splicing and expression are differentially regulated in ageing

Considering the changes in AS identified in TFCs-CpCs and ECs during ageing, we decided to first characterise and then manipulate the expression of the *Smu1* gene, a core component of the splicing machinery in *Drosophila* and itself subjected to differential GE and AS during ageing (see below). *Smu1* has three identified mRNA isoforms that share a common open reading frame (509 amino acids) but present different 5' UTRs. They all contain four mini-introns, with sizes varying between 56 and 68 bases long (Fig. 4a). The analysis of *Smu1* AS events during ageing indicated that TFCs-CpCs presented a significant retention of the second intron, with a dPSI of 18.2% at 4 weeks (Fig. 4b; Supplementary Fig. 4). The retention of this mini-intron in the coding region of *Smu1*'s mature mRNA results in a premature termination codon within the LisH dimerising domain, a domain known to interact with Beag, RED in humans[48,49]. Since Smu1/RED interaction is needed for proper splicing of small introns, at least in human cells[50], the retained intron in old Smu1 TFCs-CpCs may imply reduced levels of functional Smu1 and, as a consequence, a hindrance to small intron splicing in those cell types. In contrast, ECs did not display significant

AS changes in the *Smu1* gene during the 4-week period analysed (Fig. 4b; Supplementary Fig. 4).

Both cell groups (TFCs-CpCs and ECs) also showed distinctive patterns of *Smu1* gene expression during ageing, as determined by RNA-seq and by quantitative digital PCR. First, the amounts of *Smu1* mRNA in ECs ranged from ~100 to ~400 copies of *Smu1* mRNA/ng of cDNA and in TFCs-CpCs from ~0 to ~3. Second, while TFCs-CpCs expressed similar, low *Smu1* levels at 1- and 4-weeks, at 3-weeks it is almost undetectable. ECs, on the contrary, displayed the opposite profile, with a strong induction of *Smu1* expression at 3-weeks (Fig. 4c–e). These findings, coupled with the genome-wide RNA-seq analyses of GE and AS events, determine that the ageing process in various niche cell populations may exhibit some similarities, yet distinct variations in both GE and AS emerge during the ageing of specific niche cell types.

## *Smu1* acts in niche cells to regulate the GSC pool and cell morphology

Having determined that *Smu1* was expressed in niche cells, we next analysed whether it was required for niche function. First, we reduced *Smu1* mRNA levels in adult niche cells utilising a combination of the *tub-Gal80^{ts}* tool with several Gal4 drivers (*hedgehog (hh)*, *patched (ptc)* or *bric-a-brac 1 (bab1)*) and scored GSC numbers. Knocking down *Smu1* mRNA in TFCs + CpCs (*hh^{ts}>Smu1 RNAi*), in ECs (*ptc^{ts}>Smu1 RNAi*) or simultaneously in the three cell types (*bab1^{ts}>Smu1 RNAi + dicer*) resulted in fewer GSCs per niche (0.0001 < $P$ < 0.05; Fig. 5a, b; Supplementary Fig. 5a–f). To confirm that the observed reduction in GSC numbers was due to *Smu1* mRNA depletion, we allowed the recovery of *Smu1* levels in *ptc^{ts}>Smu1 RNAi* females by placing the flies at permissive temperature (18 °C) for the Gal80^{ts} protein for 1 week. This treatment (1 week at 29 °C upon eclosion followed by 1 week at 18 °C) rescued the drop in GSC numbers observed in *ptc^{ts}>Smu1 RNAi* niches kept for 1 week at 29 °C (Supplementary Fig. 5g). Second, we quantified the number of CpCs in *Smu1*-depleted adult niches (*bab1^{ts}>Smu1 RNAi*) and found a significant increase in CpC numbers compared to controls (4.95 CpCs/germarium on average in controls, $n = 75$; 6.22 CpCs/germarium in *bab1^{ts}>Smu1 RNAi*, $n = 54$; $P < 0.0001$; Supplementary Fig. 6a). Importantly, in spite of this increase in CpC numbers, depletion of *Smu1* in adult niches induced GSC loss (see above). Subsequently, we lowered *Smu1* mRNA levels in ECs from adult females (*ptc^{ts}>Smu1 RNAi*) and scored their numbers and cell volumes utilising a dedicated ImageJ macro described below. Control ovaries showed an average of 27.2 ECs/ germarium ($n = 37$) whereas *ptc^{ts}>Smu1 RNAi* contained 34.2 ($n = 12$, $P < 0.0001$). Similarly, the average EC cell volume was 1.28 times larger in *ptc^{ts}>Smu1 RNAi* (410 cells in 12 germaria analysed) compared with controls (1005 in 37, $P < 0.005$; Supplementary Fig. 6b, c). Third, we engineered two *Smu1* loss-of-function alleles (*Smu1^1* and *Smu1^2*) that are lethal in trans to each other or in hemizygosity and that behave similarly in our phenotypic analyses. *Smu1^1* bears an 11-base pair deletion that generates a frame shift (early stop codon) mutation. The truncated Smu1^1 protein only contains the first 22 original N-terminal amino acids followed by 6 ectopic ones. The *Smu1^2* allele contains a 12-base deletion that removes 4 amino acids (M24-K25-T26-L27) from the LISH domain but preserves the rest of the open reading frame (Fig. 5c). We generated mosaic niches containing *Smu1^1* mutant cells, including CpCs and ECs. While the number of cells forming the CpC rosette did not vary between control and mosaic niches, we found that mosaic niches contained fewer GSCs, in agreement with our results using *Smu1* RNAi ($P < 0.05$; Fig. 5d, e). The analysis of mosaic rosettes gave rise to another interesting observation, as mutant CpCs possessed a cell area 1.24 times larger than controls ($n = 23$ control cells, $n = 42$ *Smu1^1* cells, $P < 0.005$; Fig. 5f, g). With this in mind, we next used the MARCM technique to label control and *Smu1^1* ECs and to compare their cell volume. Mitotic recombination was induced at 3rd larval

instar and at the white pupa stage. We determined that *Smu1^1* ECs were 1.82 times larger ($P < 0.005$) and—assuming that the MARCM labelling was equally effective in both genotypes—divided more frequently than controls, as the number of MARCM-labelled cells per germarium increased from 1.7 in controls to 2.5 in *Smu1^1* mutants (36 control ECs in 21 germaria analysed, 57 *Smu1^1* ECs in 23; Fig. 5h, i). Overall, the above results strongly suggest essential functions for *Smu1* in niche cells to control their proliferation and morphology and to maintain a stable GSC population. While how a core splicing factor affected cell proliferation and cell shape is unclear, the fact that decreasing *Smu1* in ECs (see below) induced GE or AS changes in genes known to participate in cytoskeleton dynamics (i.e., *dysc, Mlc2, Abi, a-spec, Klp31E, mim, scra, Unc-115a* or *Zasp66*), in the control and coordination of growth (*mav*) or in the activity of small GTPases (*RhoGAP102A, CG5521, Graf* or *ric*; Supplementary Data 3) may provide an explanation for the observed phenotypes.

Finally, considering the marked increase in *Smu1* levels in 3w-old ECs, we overexpressed *Smu1* in ECs using the *ptc-Gal4* driver to try and mimic ageing conditions. In this condition, we could detect a significant decrease in the number of GSCs hosted within the niche (control, 2.28 GSCs/germ, $n = 25$; *ptc^{ts}>Smu1^{DP01419}*, 1.9 GSCs/germ, $n = 30$, $P < 0.05$; Supplementary Fig. 6d). Hence, *Smu1* overexpression in escort cells seems to recapitulate some aspects of niche ageing. Together with the loss-of-function experiments, the above findings recognise the regulation of *Smu1* as an essential factor for ovarian niche homeostasis.

## *Smu1* controls GE and AS in escort cells

With the idea of identifying potential Smu1 target genes, we utilised the *ptc^{ts}>Smu1 RNAi* combination to deplete *Smu1* and looked at GE and AS events in ECs. We selected this specific cell type because *Smu1* mRNA levels were significantly higher in ECs and because the impact of *Smu1* depletion on GSC maintenance was notably more severe in ECs compared to TFCs-CpCs (Fig. 4d, e; Supplementary Fig. 5b–f). We isolated ECs from control and *ptc^{ts}>Smu1 RNAi* ovaries and conducted bulk RNA-seq on two biological replicates for each genotype. The efficiency of the RNAi tool was confirmed by the 5-fold reduction in *Smu1* mRNA levels observed in experimental cells (Supplementary Data 3). We identified 118 genes with GE changes (fold change <0.5 or >2; false discovery rate <10%) and 210 AS events corresponding to 189 genes (dPSI > 15). The enriched GO cellular components and biological processes terms in the GE dataset identified representative changes related to membrane biology and proteostasis, whereas the AS genes grouped under cell adhesion, cell polarity, cytoskeleton and neural signalling/development (Fig. 6a–d; Supplementary Fig. 7a, b). The KEGG analysis of the AS events yielded two pathways, endocytosis and efferocytosis, represented with 7 and 5 genes, respectively, none of which showed differential GE in *ptc^{ts}>Smu1 RNAi* ECs. In fact, only 3 genes (*CG13937, Scaf* and *TM4SF*) were common to both GE and AS datasets (Fig. 6e, f; Supplementary Fig. 7c). Finally, comparing the transcriptome of *Smu1*-depleted ECs with that of aged ECs revealed that decreasing swiftly *Smu1* mRNA levels in ECs did not recapitulate extensively molecular aspects of EC ageing. We could only find 5 genes (out of the 118 differentially expressed in *Smu1*-depleted ECs) in common with the aged EC dataset (*CG40485, dpr17, TM4SF, Cyp4p2* and *Mlc2*; the former three were up-regulated in both aged and *Smu1* RNAi conditions whereas the latter two showed opposite regulation). In the case of genes showing AS changes, of the 639 identified in aged ECs and of the 189 found in *Smu1 RNAi* ECs, 53 genes were common. 35 of the latter showed AS events in the same category in both datasets (65 events), 12 of which were the same event, 8 similarly regulated (Fig. 6g, h). The CDF analysis did not give significant differences between controls and genes showing differential AS (Supplementary Fig. 7D, E). We conclude that the molecular signatures behind EC ageing and the decrease in *Smu1* activity within ECs show limited overlap.

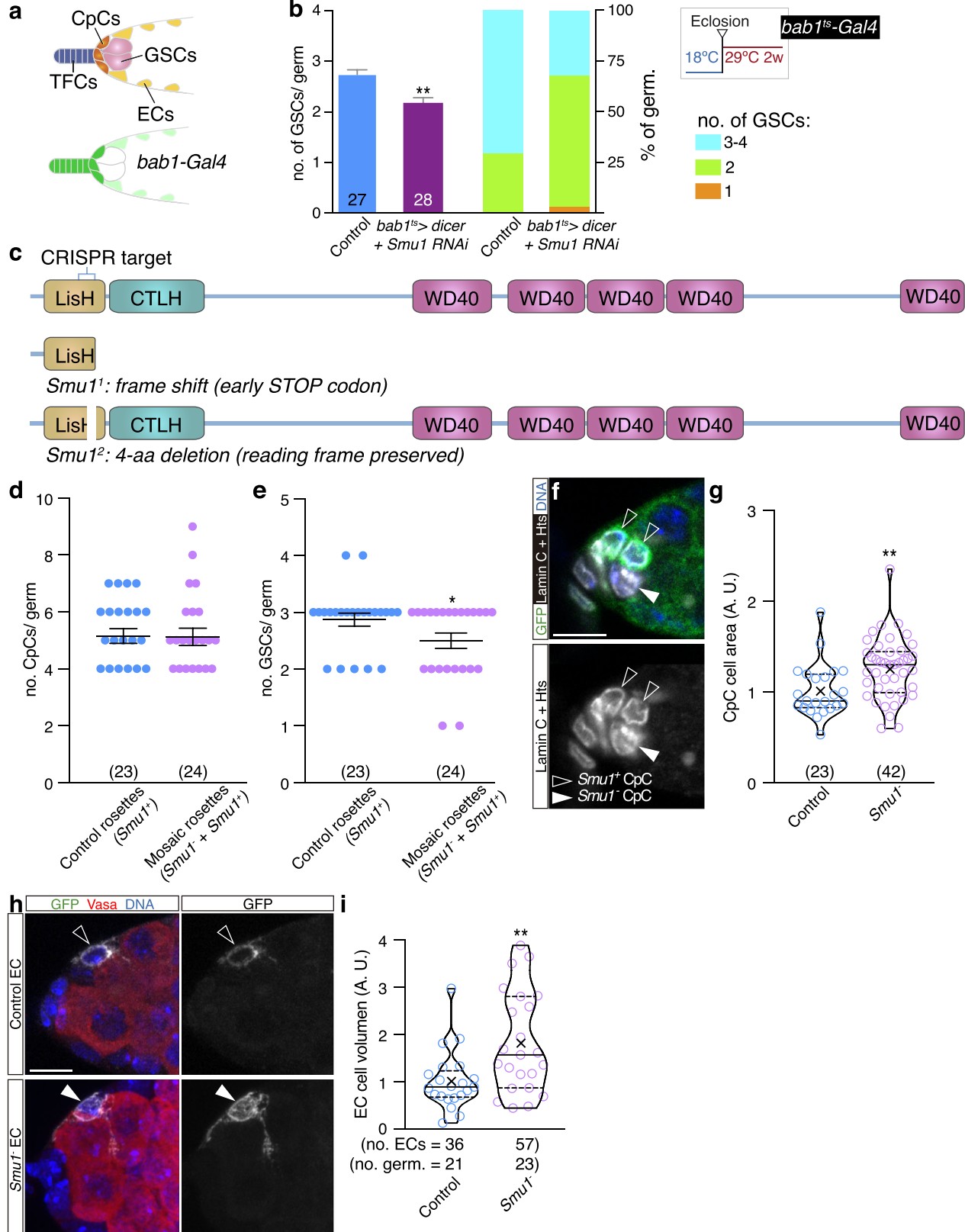

## Fas2 is essential for niche functioning and its localization reveals CpC polarity

An example of the concurrence of GE and AS changes during ageing is the *Fasciclin 2* gene, the *Drosophila* orthologue of human NCAM-1 and -2. Although best known for its role in neural synapsis, *Fas2* is also expressed in the ovary[39–41]. *Fas2* transcription gives rise to seven

different mRNAs (FlyBase r6.49; *Fas2-RA, RB, RC, RD, RF, RG*, and *RH*) that generate seven different protein isoforms, four of which−containing exons 10 to 13 and known as the Fas2 neural isoforms−are recognised by the 1D4 monoclonal antibody: PA, PD, PF and PG (Fig. 7a; Fas2-PH also contains exon 10, so it may as well be recognised by 1D4, but this remains to be determined)[48,51]. According to our RNA-seq data,

**Fig. 5 | *Smu*1 is required in niche cells for GSC maintenance and for niche cell morphology. a** The *bab1-Gal4* line is expressed strongly in TFCs-CpCs and weakly in ECs. **b** Quantification of GSC numbers and percentage of germaria containing 1, 2 or 3–4 GSCs in control and in *bab1ts>Smu1 RNAi + dicer* experimental germaria. Flies were grown at 18 °C and, upon eclosion, kept at 29 °C for two weeks prior to dissection. Numbers in the columns denote the sample size per genotype (*n*). Colour code for the number of GSCs found in germaria. **c** Scheme showing the different protein domains within Smu1, the target region for the CRIPSR editing and the molecular characterisation of the two *Smu1* alleles reported in this work. **d** The average number of CpCs found in control *(Smu1+)* and mosaic *(Smu1+ +* *Smu1)* rosettes is not significantly different. **e** However, the average number of GSCs per germarium is lower in mosaic CpC rosettes. **d, e** The mean and SEM are represented. **f** The average cell area of *Smu1-* CpCs is larger than that of control cells. Quantified in **g**. **h** The average cell volume of *Smu1-* ECs is larger than that of control cells. Quantified in **i**. In addition to the individual values, violin plots also show the first and third quartiles, the median and the mean (X). Numbers in parenthesis indicate sample size per genotype (*n*). *P* values of two-tailed, unpaired *t*-tests considered statistically significant between different genotypes are shown (**e**: *P = 0.0428, **b**: **P = 0.0032, **g**: **P = 0.0048, **i**: **P = 0.0043). Scale bar: 10 μm. Related to Supplementary Figs. 5 and 6.

*Fas2* is differentially expressed and alternatively spliced during ageing in TFCs-CpCs. Thus, in 1-week old cells, exon 4-containing isoform *RF* constituted 38.2% of transcripts including exons 3 and 5 (i.e., all of the Fas2 isoforms), while *RA* represented 41.6% of all isoforms containing exons 10 and 13. In contrast, the proportion of *RF* and *RA* transcripts changed significantly in 4-week old cells, as *RF* represented 6.9% of all transcripts and *RA* 8.1% of the mRNAs containing exons 10 and 13 (Fig. 7a; Supplementary Fig. 8). In addition to these changes in AS, *Fas2* transcription at 4 weeks was upregulated ~4-fold compared to 1w cells (Fig. 7b).

Using the anti-Fas2 1D4 monoclonal antibody, we established that Fas2 was localised to the membrane of CpCs, ECs and follicle cells and their precursors in the germarium (Fig. 7c). In order to test whether the changes in *Fas2* GE and AS during ageing had any consequences for Fas2 membrane localisation in the GSC niche, we utilised the 1D4 monoclonal antibody to quantify neural Fas2 isoform levels in CpCs from 1w- and 4w-old niches. First, we determined that 1w-old CpCs showed polarized localization of Fas2, as CpC-CpC boundaries possessed significantly lower neural Fas2 amounts than CpC-GSC contacts (*P* < 0.0005; Fig. 7c, d). We further established that CpCs exhibited cell membrane asymmetries by looking at the distribution of polarity markers such as α-Spectrin and Discs large (Dlg), which themselves accumulated differently in CpC-CpC and CpC-GSC contacts (0.0001 < *P* < 0.005; Supplementary Fig. 9a–d). Second, we confirmed that, while CpC-GSC contacts did not show significant differences in 1D4 signal between young and old niches, CpC-CpC boundaries contained significantly higher amounts of neural Fas2 in 4w-old samples (*P* < 0.0001; Fig. 7c, d). From these results we conclude that the increase in *Fas2* transcription in TFCs-CpCs during ageing is reflected in a surge of Fas2 neural isoforms in CpC-CpC contacts specifically, pointing to an age-dependent polarized distribution of Fas2 isoforms in CpCs.

Importantly for this work, it was reported that Smu1/Beag-mediated *Fas2* alternative splicing procured a critical function in larval neuromuscular synapses. Smu1 and Beag are conserved components of the *Drosophila* spliceosome[52] and a reduction of either in neuromuscular junctions resulted in a change in the distribution of Fas2 neural isoforms and in fewer and larger synaptic boutons[48]. Thus, synaptic morphology and function depended on the correct splicing of the *Fas2* gene. To test if *Smu1* activity in the niche could affect Fas2 localization in CpCs, we analysed 1D4 distribution in *Smu1*-depleted CpCs (*hhts>Smu1 RNAi*) and found that the accumulation of neural Fas2 in CpC-GSC boundaries was reduced compared to controls (*P* < 0.0001). However, the localization of 1D4 to CpC-CpC contacts was not affected by the decrease in *Smu1* levels (Fig. 7e, f). Of interest, *hhts>Smu1 RNAi* CpCs compromised neither α-Spectrin nor Dlg distribution (Supplementary Fig. 9b, d), demonstrating that *Smu1* activity specifically disturbed neural Fas2 accumulation and that it did not affect the overall polarization of these niche cells. Therefore, we surmise that *Smu1* function in CpCs might affect *Fas2* splicing and thus the polarized distribution of some of the isoforms.

Finally, we tested whether Fas2 was required for proper niche functioning. We first decreased *Fas2* levels using RNA interference (*hhts>Fas2 RNAi*) in CpCs and found a significant reduction in GSC numbers in experimental niches (*P* < 0.0001; Fig. 7g). Second, we made use of the EP1462 line to overexpress *Fas2* in CpCs and quantify GSCs. We found a significant increase in GSC numbers upon *Fas2* over-expression (control, 2.63 GSCs/germ, *n* = 30; *hhts>FasEP1462*: 3.12 GSCs/germ, *n* = 33; *P* = 0.0047; Fig. 7h). While this is a somehow unexpected result (the observed rise in *Fas2* levels in 4w-old CpCs correlates with GSC loss), the interpretation of the gain in GSC numbers in *hhts>FasEP1462* is complex, as with the EP line we allegedly induced expression of all *Fas2* isoforms and this may be different to the real scenario in aged cells, in which isoform expression is heterogeneous. As a control for *Fas2* overexpression, we confirmed that Fas2 levels were augmented in *hhts>FasEP1462* (Supplementary Fig. 9e). Hence, while the extent to which the enrichment of neural Fas2 in CpC-CpC boundaries in 4w-old germaria influences GSC loss remains unclear, our results directly imply Fas2 in the maintenance of the GSC pool.

## Discussion

We find a number of transcriptomic changes during the ageing of somatic support cells of the ovarian germline niche. These include not only fluctuations in gene expression but also in the patterns of alternative splicing, including exon usage, alternative splicing sites and intron retention events. The alterations observed in GE and AS primarily show cell-type specificity, given the distinct molecular signatures associated with the ageing of TFCs-CpCs and of ECs. While this might imply specific ageing programmes, some events are common to all cell types. For instance, 39 differentially expressed genes (out of a total of 122; 31.9%) in aged ECs show a similar variation in TFCs-CpCs and most of these are related to cytoskeleton remodelling, cell shape and muscle organization. In the case of AS the situation is even more restricted, as we could identify only 33 identical events (corresponding to 28 genes out of 639; 4.38%) happening in both cell types during ageing, again with an enrichment of genes involved in cytoskeleton and muscle activity.

In addition to muscle development, organization and activity, the functions of the genes falling within the muscle category in these cell types also relate to actin dynamics and to cell shape, as some of them bind or colocalize with actin (i.e., *Mlp60A, Mp20, Zasp66, Tm1* or *Vinc*) or code for different actins (such as *Act57B*). Since the loss during ageing of actin cytoskeleton organization and dynamics is well documented[53], we suggest that the common, changing molecular processes identified in aged niche cells may affect niche stiffness, intercellular communication—one of the general hallmarks of ageing— and/or niche cell morphology, factors directly implicated in effective niche-stem cell signalling and that can be altered during ageing[2,54,55]. A gradual decline in the efficiency of intercellular communication may explain the observed decrease of GSC numbers in 4-week old ovaries, as niche cell-GSC interaction relies on cellular projections sent by GSCs, CpCs and ECs[56–58]. Of interest, other stem cell systems also utilise thin cellular projections for communication such as the male germline niche or the lymph gland in *Drosophila*[59,60] and nanotubes have been implicated in cancer stem cell niche formation[61]. It would be interesting to study if intercellular communication mediated by thin cellular projections in these systems is also affected by ageing.

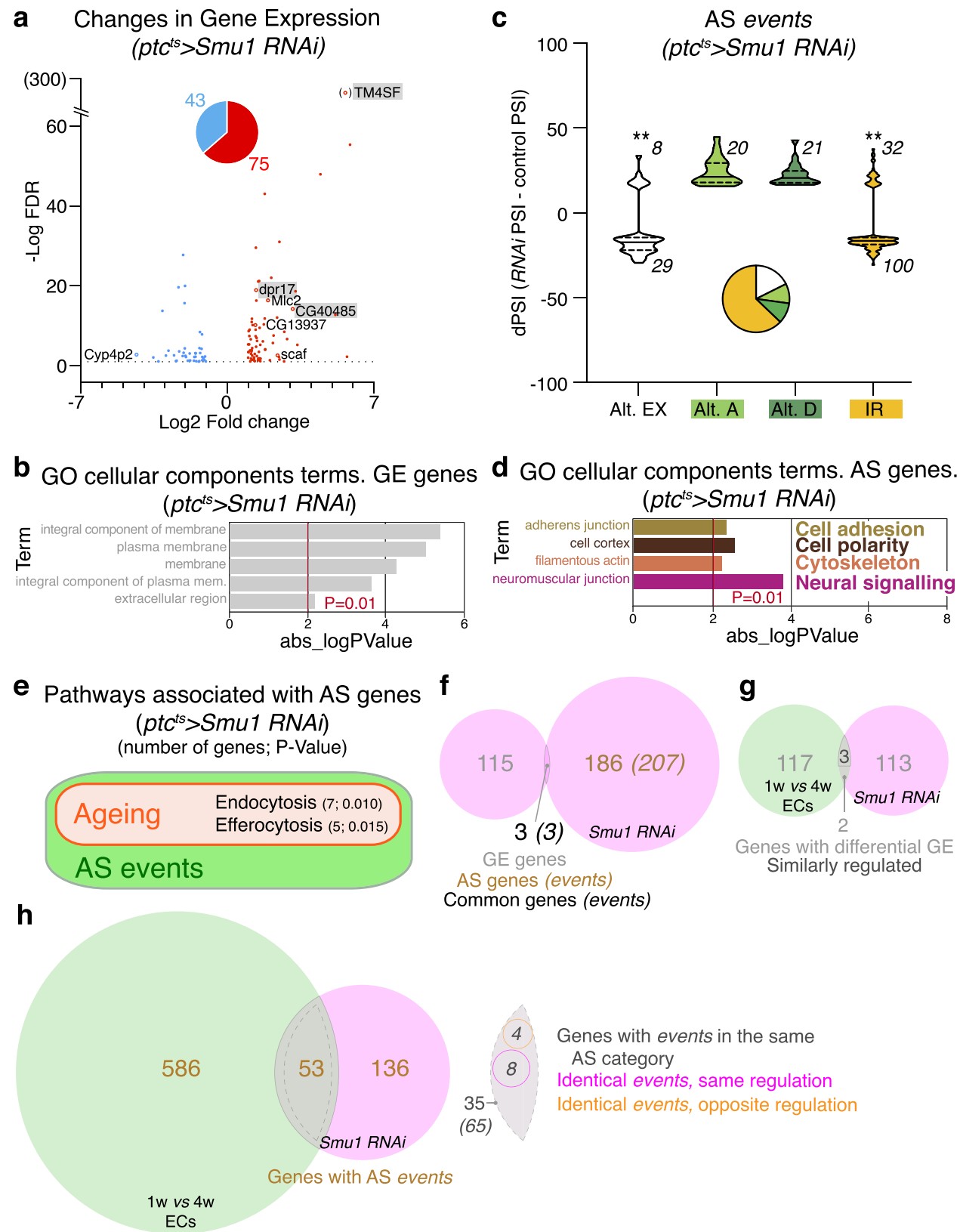

A recurrent category detected in our analyses is that of neural signalling/development. The GO terms comprised in this group include those related to axonal growth and projections, synapse assembly, signalling and nervous system development and morphogenesis. While the presence of cellular projections in cap cells and escort cells might explain some of the identified categories, our findings strongly suggest that the ovarian niche cells share some similarities with neural cells. Additional validation of our results comes from published transcriptomic analyses by bulk and single-cell RNA-seq, which report expression of genes associated to nervous system development, neural activity genes, axon growth and synaptic target transmission in GSC niche cells[41,46,62]. The presence of ion channel

**Fig. 6 | Differential GE and AS events in *Smu1*-depleted escort cells. a** Changes−represented as log2 fold change of *ptc^ts^>Smu1 RNAi*/control *versus* the -log10 FDR−in GE identified 75 genes upregulated and 43 genes downregulated in *Smu1 RNAi* compared to controls (linear fold change <0.5 or >2; FDR < 10%). **b** Terms identified after a GO-cellular components analysis of GE changes. **c** Distribution of the 210 changes in AS events in *Smu1 RNAi* compared to controls. Changes are represented as variations in percent splicing inclusion (dPSI = 4w PSI − 1w PSI). Only events with an absolute dPSI>15 were considered. Statistically significant *P* values of the compare proportions test are shown (**P < 0.001). Pie chart represents the different AS categories. **d** Terms identified after a GO-cellular components analysis of AS genes. **e** Among the genes with changes in AS, genes associated to two pathways previously shown to be related to ageing, endocytosis and efferocytosis showed significant enrichment. None of the twelve genes associated to the latter pathways were differentially expressed in *ptc^ts^>Smu1 RNAi* ECs. **f** Three genes showed changes in both GE and AS in *ptc^ts^>Smu1 RNAi* ECs (*CG13937, TM4SF* and *Scaf*). **g** Diagram representing the number of differentially expressed genes in *ptc^ts^>Smu1 RNAi* and 1w *versus* 4w ECs datasets. Of the 5 genes common to both, 3 showed the same GE variation (either up- or down-regulated). **h** Diagram to indicate the number of genes showing AS changes in *ptc^ts^>Smu1 RNAi* and 1w *versus* 4w ECs datasets. Of the 53 common genes, 35 (corresponding to 65 events; in italics) share events in the same AS category (Alt EX, Alt A, Alt D or IR). Of the 65 events, 12 (each one corresponding to one gene) are identical in both groups, 8 of them with the same kind of dPSI variation (i.e., a positive or a negative dPSI; the 8 events are 3 Alt EX, 1 Alt D and 4 IR) and 4 with opposite variation (1 Alt EX and 3 IR). Data correspond to two biological replicates per genotype. Related to Supplementary Fig. 7, Supplementary Data 3 and Supplementary Data 4.

components and calcium signalling in niche cells corroborates this view and strengthens previous findings demonstrating that escort cells receive the neurotransmitter octopamine to promote GSC proliferation after mating in the female ovarian niche[63]. Further investigation is needed to understand how the expression of voltage-activated ion channels, ion transport systems and calcium-signalling components is regulated in non-innervated cells such as the TFCs-CpCs, and how they influence niche function.

Considering the complex morphology of adult ECs, their dynamic cellular protrusions that extend between the germline cells, that they undergo limited replenishment and the fact that there are at least three different populations of ECs described in the germarium[31,41,64], the number of genes whose transcription was altered with age in ECs is unexpectedly low, as we detected only 122 genes with altered expression. This is in stark contrast to the TFCs-CpCs results, which identified 1400+ genes with GE changes. Since both groups present a similar transcriptomic complexity −our RNA-seq analyses identified 10,400+ genes in TFCs-CpCs and 12,000+ in ECs−one explanation for the above finding is that both cell groups present different sensitivity to local and/or systemic variations such as ageing, with TFCs-CpCs exhibiting a particularly refined regulation of transcription. Leveraging the known function of cap cells in the production of niche signals including Hedgehog, Dpp (the fly BMP2/4 homologue), Unpaired (a Jack/STAT ligand) or Wnt, and their dependence on general conditions such as nutrition[65,66], our results suggest these cells display high transcriptional dynamics, enabling them to adapt promptly to subtle changes in their physiological context. Alternatively or in addition, ECs may be under selective pressure to limit their variability during ageing, as significant transcriptional alterations could potentially have severe implications for initial germline differentiation, which takes place in close contact with ECs and may require more stringent physical restrictions. Thus, the fact that there are only 122 genes represented in the GE group in ECs is an indication of their robustness. A third possibility may imply that EC populations−which, contrary to TFCs and CpCs, are replenished during adult life−age at a different rate than TFCs-CpCs, as shown for other cell types in *Drosophila* adults[67]. In our model, transcription can be finely regulated, as there are feedback loops that correct for variations in transcription and they are in place in ECs because of their importance. Changes in AS, on the other hand, may represent a more terminal phenotype that cannot be as efficiently buffered by feedback loops, even though some splicing genes are themselves subjected to alternative splicing (i.e., *mbl* or *Smu1*). Thus, the similar numbers of AS events in aged TFCs-CpCs and ECs may reflect cell-type specific impairment of the co-transcriptional mechanisms governing pre-mRNA processing, as alternative splicing decisions are made co-transcriptionally, coupled to transcription elongation[68]. Importantly, an increase of the average transcriptional elongation speed and changes in splicing are common ageing features across different organisms, including fruit flies[28].

Smu1 and its partner Beag (RED in humans) are general splicing factors in humans and in flies. Recently, Smu1 and RED have been also reported as alternative splicing regulators, primarily involved in the efficient processing of short introns by relieving steric constrains posed by short 5'-3' intron distances[48,50]. Our finding that *Smu1* expression is regulated during ageing suggests that general splicing and/or alternative splicing changes with age in the ovarian niche cells. In fact, we demonstrate that GE and AS of alternative splicing factors varies during ageing, adding further evidence of a role for AS during stem cell niche ageing. We have identified 15 splicing factors that show changes in either GE (*bru1, bru2* and *CG11360*), in AS (*Doa, CG4119, CG4896, mub, ps, Psi, Saf-B, Sxl* and *tra2*) or in both (*Smu1, bru3* and *mbl*) in TFCs-CpCs during ageing. Moreover, our *Smu1* knockdown analysis in ECs identified 118 genes with GE changes and 189 genes (corresponding to 210 events) with differential AS. Interestingly, among those we could not find *Fas2*, a known target of Smu1/Beag splicing activity in larval neuromuscular junctions[48], indicating that *Smu1*-mediated AS is cell specific. Finally, while removing *Smu1* function from CpCs or ECs causes GSC loss and an increase in cell size, indicating that *Smu1*-dependent splicing has an essential function during ovarian niche homeostasis, its role during ageing is unclear, as decreasing *Smu1* levels in ECs does not elicit significant molecular prints of ageing.

We set out to determine if ageing impacts alternative splicing within the support cell populations of a well-defined stem cell niche. We have unveiled that ageing affects both gene expression and the occurrence of AS events in a cell-type-manner. Moreover, a significant proportion of genes within TFCs-CpCs orchestrate concurrent changes in GE and AS. While acknowledging that ageing can disrupt transcription elongation, leading to general mRNA maturation dysregulation, including splicing[28,68], our results imply the existence of cell-type specific ageing signatures dictating distinct molecular outcomes in the fly's ovarian niche and confirms the observation that different cell types may age with specific ageing features[67]. The universality of our discoveries across other adult stem cell niches experiencing ageing necessitates further investigations. Nonetheless, we hypothesize that the regulation of alternative splicing in support cells, potentially combined with differential gene expression, contributes to stem cell loss due to niche ageing.

## Methods
### Fly stocks
Flies were grown at 18 °C, 25 °C or 29 °C on standard medium. The lines used include:

*y w* (RRID:BDSC_6598)

*bab1-Gal4: w;; bab1-Gal4/TM3*[69].

*tau::GFP: w;; UASp-tau::mGFP6*[70].

*mbl::Gal4: w; PBac{w[+mC]=IT.GAL4}mbl^0217-G4^* (RRID:BDSC_77534).

*hh-Gal4: y[1] w[*];; wg[Sp-1]/CyO; P{w[+m*]=GAL4}hh[Gal4]/TM6B, Tb[1]* (RRDI:BDSC_600186).

*ptc-Gal4: w[1118];; P{y[+t7.7] w[+mC]=GMR69F10-GAL4}attP2* (RRID:BDSC_45900).

*ptc-LexA: w[1118]; P{y[+t7.7] w[+mC]=GMR69F10-lexA}attP40* (RRID:BDSC_54926).

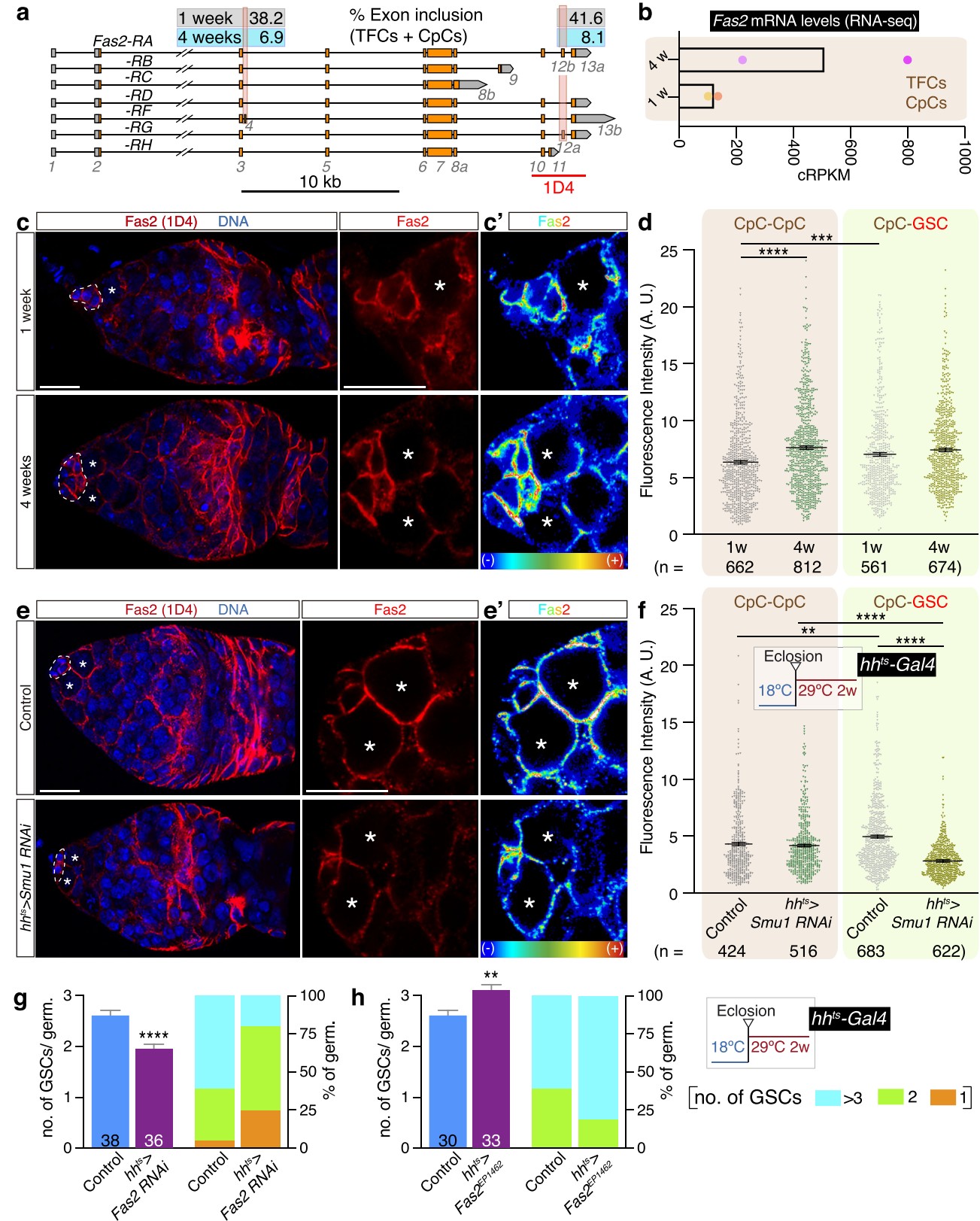

LexAop-mCD8::GFP: y[1] w[*] P{y[+t7.7] w[+mC]=13XLexAop2-mCD8::GFP}su(Hw)attP8 (RRID:BDSC_32204).

UAS-Fas2 RNAi: y sc v sev[21];; P{y[+t7.7] v[+t1.8]=TRiP.HMS01098}attP2 (RRID:BDSC_34084).

UAS-Fas2^EP1462: w[1118] P{w[+mC]=EP}Fas2[EP1462] (RRID:BDSC_11231).

UAS-Smu1 RNAi: y sc v sev[21];; P{y[+t7.7] v[+t1.8]=TRiP.HMS01075}attP2 (RRID:BDSC_33739).

UAS-Smu1^DP01419: y[1] w[67c23];; P{y[+t7.7]=Mae-UAS.6.11}DP01419/TM3, Sb[1] (RRID:BDSC_22072).

UAS-dicer2 (II): w[1118]; P{w[+mC]=UAS-Dcr-2.D}2 (RRID:BDSC_24650).

**Fig. 7 | An essential function for *Fas2* in the GSC niche during ageing. The polarized localization of Fas2 in CpCs depends on age and *Smu1*. a** Genomic organization of the *Fas2* gene. Percentage of exon inclusion of exons 4 and 12a is shown for 1-week and 4-week old TFCs + CpCs. **b** Quantification of *Fas2* mRNA levels with RNA-seq in TFCs + CpCs, represented as cRPKM (corrected reads per kilobase of target transcript sequence per million of total reads). Data correspond to two technical replicates of two biological replicates per time point. Following editorial guidelines, statistics such as error bars or *P* values cannot be derived from fewer than three values. **c** Germaria showing the distribution of Fas2 (isoforms **a, d, f, g** and **h**, as detected with the 1D4 monoclonal antibody; red) and DNA (blue) in 1- and 4-week old germaria. **d** Quantification of Fas2 fluorescence intensity of CpC-CpC and CpC-GSC contacts in 1- and 4-week old samples. **e** Distribution of neural Fas2 isoforms in control and *hh^{ts}>Smu1 RNAi* CpCs. (c' and e') are false-coloured images representing Fas2 signal intensity. **f** Quantification of Fas2 fluorescence intensity of CpC-CpC and CpC-GSC contacts in control and *hh^{ts}>Smu1 RNAi* CpCs. **g** Quantification of GSC numbers and percentage of germaria containing 1, 2 or 3-4 GSCs in control and in *hh^{ts}>Fas2 RNAi* experimental germaria. **h** Quantification of GSC numbers and percentage of germaria containing 1, 2 or >3 GSCs in control and in *hh^{ts}>Fas2^{EP1462}* experimental germaria. In the *RNAi* and *Fas2* overexpression experiments, flies were grown at 18 °C and placed at 29 °C for two weeks prior to dissection. Error bars indicate SEM. Data in (d, f) correspond to measurements obtained from 70<contacts<136 in 21<germaria<34 (*n* reflects the number of measurements taken for each condition). *P* values of two-tailed, unpaired *t*-tests considered statistically significant between different genotypes are shown (**f**: \*\**P* = 0.00059, **h**: \*\**P* = 0.00469, **d**: \*\*\**P* = 0.000296, **d**: \*\*\*\**P* = 1.9776E −010, **f** RNAi *versus* RNAi: \*\*\*\**P* = 9.3931E−28, **f** control *versus* RNAi: \*\*\*\**P* = 9.66E−50, **g**: \*\*\*\**P* = 0.000091). Scale bar: 10 μm. Related to Supplementary Figs. 8 and 9 and Supplementary Data 1.

*tub-Gal80^{ts}* (II): *w[\*]; P{w[+mC]=tubP-GAL80[ts]}20; TM2/TM6B, Tb[1]* (RRID:BDSC_7019).

*tub-Gal80^{ts}* (III): *w[\*];; P{w[+mC]=tubP-GAL80[ts]}2/TM2* (RRID:BDSC_7017).

*FRT Smu1^1*: *w[\*]; P{ry[+t7.2]=neoFRT}82B Smu1^1* (this work).

MARCM: *y[1] w[\*];; P{ry[+t7.2]=neoFRT}82B P{w[+mC]=tubP-GAL80}LL3* (RRDI:BDSC_5135).

*y[1] w[\*]; P{w[+mC]=UAS-mCD8::GFP.L}LL5, P{UAS-mCD8::GFP.L}2* (RRDI:BDSC_5137).

*y[1] w[\*]; P{Act5C-GAL4-w}E1/CyO* (RRDI:BDSC_25374).

*P{ry[+t7.2]=hsFLP}1, w[1118]; Adv[1]/CyO* (RRDI:BDSC_6).

*P{ry[+t7.2]=hsFLP}12, P{w[+mC]=UAS-GFP.U}1, y[1] w[\*]; P{w[+mC] =tubP-GAL4}LL7 P{ry[+t7.2]=neoFRT}82B P{w[+mC]=tubP-GAL80}LL3/ TM6C, Sb[1] Tb[1]* (RRDI:BDSC_86311).

To control the expression of different transgenes in adult niches we used the *tub-Gal80^{ts}* system. Keeping the adults at 18 °C blocks Gal4-induced gene expression, whereas placing the flies at 29 °C brings about Gal4-mediated gene expression. We used the *UAS-dicer2* construct to enhance the RNAi phenotype. To induce *Smu1^1* mutant CpCs, flies of the appropriate genotype were grown at 25 °C. To generate MARCM (Mosaic Analysis with a repressible Cell Marker) *Smu1^1* mutant clones, flies were grown at 18 °C and heat-shocked for 1 h at 37 °C both at third larval instar and at white pupae. We performed a series of tests to make sure the expression of *tau::mGFP6* did not affect niche activity. First, flies expressing *tau::GFP* with any of the Gal4 drivers utilized were perfectly viable, healthy and fertile and possessed normal-looking ovaries. In fact, flies containing both the driver and the *UASp-tau::GFP* construct were kept as stocks. Second, we scored GSC numbers and Fas2 localisation in CpC-CpC contacts of *bab1-Gal4*, *hh-Gal4*, *bab1>tau::GFP* and *hh>tau::GFP* 4-week-old flies and could not detect significant differences between them. Thus, *tau::GFP* expression in 4 week-old niche cells does not impact their function, at least as measured by GSC numbers and 1D4 levels.

## Experimental genotypes
Figure 1
(B) *y w*
(C-E) *bab1>tau::GFP: bab1-Gal4/UASp-tau::mGFP6*
Figure 2
(B-E) *bab1>tau::GFP: bab1-Gal4/UASp-tau::mGFP6*
Figure 3
*y w lexAop-mCD8::GFP/w; ptc-LexA/+*
Figure 4
(B-E) *bab1>tau::GFP: bab1-Gal4/UASp-tau::mGFP6* (TFCs-CpCs)
*y w lexAop-mCD8::GFP/w; ptc-LexA/+* (ECs)
Figure 5
(B) Control: *UAS-dicer2/+; UAS-Smu1 RNAi/TM2*
*bab1^{ts}>dicer + Smu1 RNAi: UAS-dicer2/+; UAS-Smu1 RNAi/bab1-Gal4, tub-gal80^{ts}*

(D-G) *Smu1^1* clones: *UASt-FLP/+; FRT-82B Smu1^1/bab1-Gal4 FRT-82B Ubi-GFP.*

(H, I) *Smu1^1* MARCAM: *hsFLP-1, UAS-mCD8::GFP/+; Act5C-GAL4/+; FRT-82B tubP-GAL80/FRT-82B Smu1^1*

Figure 6
Control: *13× LexAop-mCD8::GFP/w; ptc-LexA/tub-Gal80^{ts}; ptc-Gal4/+*
*ptc^{ts}>Smu1 RNAi: 13× LexAop-mCD8::GFP/w; ptc-LexA/tub-Gal80^{ts}; ptc-Gal4/UAS-Smu1 RNAi*

Figure 7
(A, B) *bab1>tau::GFP: bab1-Gal4/UASp-tau::mGFP6*
(C, D) *y w*
(E, F) Control: *tub-gal80^{ts}/+; UAS-Smu1 RNAi/TM2*
*hh^{ts}>Smu1 RNAi: tub-gal80^{ts}/+; UAS-Smu1 RNAi/hh-gal4*
(G) Control: *tub-gal80^{ts}/+; UAS-Fas2 RNAi/TM6B*
*hh^{ts}>Fas2 RNAi: tub-gal80^{ts}/+; UAS-Fas2 RNAi/hh-gal4*
Supplementary Fig. 1
(A) *bab1>tau::GFP: bab1-Gal4/UASp-tau::mGFP6*
(B) Control: *y w*
(C) *mbl>tau::GFP: mbl^{0217-G4}/UASp-tau::mGFP6*
Supplementary Fig. 2
*bab1>tau::GFP: bab1-Gal4/UASp-tau::mGFP6*
Supplementary Fig. 3
*y w lexAop-mCD8::GFP/w; ptc-LexA/+*
Supplementary Fig. 5
(A'-A") *ptc^{ts}>Smu1 RNAi: tub-Gal80^{ts}/+; ptc-Gal4/UAS-Smu1 RNAi*
(B) Control: *UAS-dicer2/+; UAS-Smu1 RNAi/TM2*
*bab1^{ts}>dicer + Smu1 RNAi: UAS-dicer2/+; UAS-Smu1 RNAi/bab1-Gal4, tub-gal80^{ts}*
(C, D) Control: *tub-gal80^{ts}/+; UAS-Smu1 RNAi/TM2*
*hh^{ts}>Smu1 RNAi: tub-gal80^{ts}/+; UAS-Smu1 RNAi/hh-gal4*
(E-G) Control: *tub-gal80^{ts}/+; ptc-Gal4/+*
*ptc^{ts}>Smu1 RNAi: tub-Gal80^{ts}/+; ptc-Gal4/UAS-Smu1 RNAi*
Supplementary Fig. 6
(A) Control: *UAS-Smu1 RNAi/TM2*
*bab1^{ts}>Smu1 RNAi: UAS-Smu1 RNAi/bab1-Gal4, tub-gal80^{ts}*
(B, C) Control: *13× LexAop-mCD8::GFP/w; ptc-LexA/tub-Gal80^{ts}; ptc-Gal4/+*
*ptc^{ts}>Smu1 RNAi: 13× LexAop-mCD8::GFP/w; ptc-LexA/tub-Gal80^{ts}; ptc-Gal4/UAS-Smu1 RNAi*
(D) Control: *y w/w; tub- Gal80^{ts}/+; P{y[+t7.7]=Mae-UAS.6.11} DP01419/TM6B*
*ptc^{ts}>Smu1^{DP01419}: y w/w; tub- Gal80^{ts}/+; P{y[+t7.7]=Mae-UAS.6.11} DP01419/ptc-Gal4*
Supplementary Fig. 7
Control: *13× LexAop-mCD8::GFP/w; ptc-LexA/tub-Gal80^{ts}; ptc-Gal4/+*
*ptc^{ts}>Smu1 RNAi: 13× LexAop-mCD8::GFP/w; ptc-LexA/tub-Gal80^{ts}; ptc-Gal4/UAS-Smu1 RNAi*
Supplementary Fig. 9

(A-D) Control: *tub-gal80^ts/+; UAS-Smu1 RNAi/TM2*
*hh^ts>Smu1 RNAi: tub-gal80^ts/+; UAS-Smu1 RNAi/hh-gal4*
(E) Control: *w Fas2^EP1462/w; tub-gal80^ts/+; TM6/+*
*hh^ts> Fas2^EP1462: w Fas2^EP1462/w; tub-gal80^ts/+; hh-gal4/+*

## Generation of *Smu1¹* and *Smu1²* alleles

Both alleles were generated using CRISPR. A stable UAS-guide RNA line was obtained with the following target sequence: ACGTTTGCAG GGTCTTCATA. *Smu1¹* bears an 11-base pair deletion (…AGCAGTA TCTGAAGGAGTCCAACC-11 deleted nucleotides-CTGCAAACGTTACA GGTTAGTACTT…) that gives rise to an early stop codon (shown in red). *Smu1²* contains a 12-base pair deletion (…GCAGTATCTGAAGGA GTCCAACCT-12 deleted nucleotides-GCAAACGTTACAGGTTAGTAC TTG…). The primers used for sequencing genomic *Smu1* DNA are (F) ATCGACCAACACCACTCAGTT and (R) GACACCCGTTTCCTCCTGAA.

## Immunohistochemistry

Adult ovaries were dissected in PBT (phosphate buffered saline +0.1% Tween 20) from yeasted females. Stainings were performed at room temperature and following standard procedures. Incubation with primary antibodies was performed overnight at the following concentrations: mouse anti-hts (1B1, Developmental Studies Hybridoma Bank (DSHB), RRID:AB_528070), 1:100; mouse anti-LaminC (LC28.26, DSHB, RRID:AB_528339), 1:30; rabbit polyclonal anti-Vasa (a gift from H. Jäckle), 1:4000; guinea pig polyclonal anti-MSP300 (a gift from T. Volk), 1/300; alpaca anti-GFP-Booster_Atto488 (ChromoTek Cat# gba488-100, RRID:AB_2631386), 1:200; mouse anti-Fasciclin II (1D4, DSHB, RRID:AB_528235), 1:100; mouse anti-Discs large (4F3, DSHB, RRID:AB_528203), 1:20; rabbit polyclonal anti-α-Spec (a gift from R. Dubreuil), 1:500. Secondary antibodies FITC, Cy2, Cy3 and Cy5 (Jackson Laboratories; final concentrations of 1:100) were incubated for four hours. To stain DNA, ovaries were incubated for 10 min with Hoechst (Sigma, 5 mg/ml; 1:1000 in PBT). To visualise actin filaments, samples were incubated 20 min in PBT + 1:20 Rhodamine-phalloidin.

## Image analysis

Confocal Z stacks of fixed samples were taken at 0.5 µm intervals using a 63×/1.4 NA oil immersion objective. Colour depth was set to 8-bit and configured so that most pixels were within the range of detection. Images were analysed utilising the Imaris or Fiji[71] softwares, and processed with Adobe Photoshop and Adobe Illustrator. Average fluorescence intensity per pixel was quantified with the Fiji "Region of Interest manager" analysis tool. We drew small squares (of ~0.1–0.5 microns² each) per germarium that contained regions of bright signal. Average background levels were measured in each germarium with at least 5 boxes in GSC nuclei or cytoplasm.

## Quantification of cell volumes and areas

Cellular volumes of ECs were measured using a Macro created in FIJI in which the GFP signal from confocal Z stacks of fixed samples taken at 0.7 µm intervals was subjected to outlier's removal ("Remove Outliers…", "radius=5 threshold=20 which=Dark stack"). General background was subtracted ("Subtract Background…", "rolling=30 sliding stack") and the resulting image was smoothened using convolution with a Gaussian function ("Gaussian Blur…", "sigma=1.5 stack"). Image intensity was processed applying a non-linear contrast stretching ("Gamma…", "value = 1.10"). Intensity threshold was manually determined considering the brightness and contrast of the images (setThreshold(adjusted value, 255)). Finally, background was set to black (setOption("BlackBackground", true)) and image was converted to mask ("Convert to Mask", "background=Dark black"). This mask was used to quantify cell volumes employing ("3D Geometrical Measure"). CpC cell areas were measured with the ROI Manager tool in FIJI after selecting the z-section containing the largest cell area.

## Isolation of niche cells by Fluorescence-Activated Cell Sorting (FACS)

Preparation of single-cell suspensions were done following the protocol by Lobo-Pecellin et al. [15]. Approx. 120 ovary pairs were dissected in Schneider's insect medium (Biowest L0207-500) supplemented with 10% (v/v) heat-inactivated foetal bovine serum (Gibco 10500-064; S-FBS), teased apart manually with dissecting forceps, washed twice with Cell Dissociation Solution Non-enzymatic 1× (Sigma C5914) and incubated with the digestion solution (670 µl Trypsin 1X [SIGMA T4299] + 330 µl Collagenase [SIGMA C9407] 10 mg/ml in PBS + 100 µ DNASe I [Sigma AMPD1-1KT] + 100 µl DNASe I buffer [Sigma AMPD1-1KT]) at 25 °C for 15 min, rolling at 40 rpm in a dissecting well. Samples were then mechanically disaggregated in a blue pipette tip and transferred to a FACS tube (BD Falcon 352054). 1 ml S-FBS and 100 µl Stop Solution were added to inactivate trypsin and DNAse, respectively. Cells were re-suspended in 1 ml S-FBS after filtering the supernatant through a 70 µm nylon mesh followed by centrifugation at 420 × g for 5 min (4 °C). The resulting cell suspension was filtered again through a 70 µm nylon mesh. Propidium iodide was used as a cell viability dye (50 µg/ml per million cells).

To isolate *bab1>tau::GFP* terminal filament and cap cells, the suspended cells were separated according to cell size (forward and side scatter plots), negative propidium iodide label and high GFP-signal intensity. *LexAop-mCD8::GFP; ptc-LexA* escort cells were selected as propidium iodide-negative, GFP-positive cells. In both cases, we used a FACSAria cell sorter (BD) equipped with a 100 µm nozzle and at low pressure (20 psi). Cells were sorted directly into 0.5 ml lysis buffer and snap-frozen in liquid nitrogen. Typically, we could sort ~1000 TFCs-CpCs or ~35,000 ECs per experiment. As a control for cell purity and germline contamination in the sorted cell populations, we confirmed that the germline-specific gene *vasa* was not present among the 10,491 genes detected in TFCs-CpCs, the 12,172 in ECs or the 11,783 in the *Smu1* RNAi experiment. To assess cell purity of the TFC-CpC and EC samples, we confirmed that Lamin-C levels (Lamin-C is a marker expressed at higher concentrations in TFCs-CpCs than in ECs[30]) are at least 17× (1 week) or 33× (4 weeks) higher in TFCs-CpCs than in ECs in our datasets. Similarly, *engrailed* (a gene that is highly enriched in TFCs-CpCs but not in ECs[69,72]) levels are 22× (1 week) or 102× (4 weeks) higher in TFCs-CpCs compared to ECs of the same ages. As for EC markers, we have compared the top 50 genes expressed in each of the three types of ECs (anterior, central and posterior) reported in ref. [41] with our TFCs-CpCs and ECs datasets. We have identified 3 genes (*CG17321*, *CG6067* and *SPE*) expressed in both the Rust et al. and our EC lists but not in our TFCs-CpCs dataset. From all these analyses we conclude that there is no significant contamination of ECs in the TFC-CpC experiment and vice versa.

## Preparation of transcriptomic samples

RNA from the different biological replicas was isolated using RNeasy Mini kit (QIAGEN). cDNA synthesis, library preparation and amplification (Pico Profiling) are described elsewhere[73]. In short, after reverse-transcription, each cDNA sample was added to an amplification mix that was subdivided into five equivalent parts for PCR amplification (26 cycles). Amplified cDNA collections were purified with the PureLink PCR Purification Kit (ThermoFisher Scientific), resuspended in 40 µl and their concentrations measured using a Nanodrop 1000 spectrophotometer.

## Bulk RNA-seq and bioinformatic analyses

The quality of total RNA isolated from sorted cells was checked using Bioanalyzer (Agilent). We prepared and sequenced twelve strand-specific Illumina libraries. On average, 80 million 125-nt paired-end reads were generated for each of the samples. Differentially expressed genes were identified using DESeq (v. 1.40.2.)[74]. Only gene expression changes with a 2-fold difference and false Discovery Rate (FDR) $p$-value ≤ 0.05 were considered. Alternative Splicing analyses were

carried out using vast-tools v2.5.1 for dm6 (VASTDB library: vastdb.dme.23.06.20.tar.gz)[23,44,75]. Differentially regulated AS events were identified with vast-tools compare and they required (i) an average ΔPSI (percentage spliced in) between samples of at least 15%, and (ii) a ΔPSI between the ranges of both samples of at least 5. To avoid underestimating the correlation of genes with differential expression in the sample pairs (1 w *versus* 4w or controls *versus* experimental RNAi) GSEA (Gene Set Enrichment Analyses) were conducted with the PANGEA tool[76] (Supplementary Data 4). Pathway enrichment analyses were done at the KEGG pathway database (https://www.genome.jp/kegg/pathway.html). GO term analyses were performed with a DAVID functional annotation tool (https://david.ncifcrf.gov) and *P*-values were calculated with the modified Fisher's Exact test EASE. Significant differences were set at $P \leq 0.01$.

## Quantification of mRNA levels by droplet digital PCR
Pico profiled samples were used to quantify *Smu1* mRNA levels by droplet-digital PCR (ddPCR). Primers used were (5′-3′): *RpL32* (used to normalise the relative amounts of mRNA), F: ATGACCATCCGCCCAG-CATAC, R: GCTTAGCATATCGATCCGACTGG; *Smu1* F: GTCCATA-GAAATCGAATCAGCG, R: ACCGTATTCAGGGAGACACC. Each reaction contained 10 µl of Master Mix ddPCR EVAGreen (Bio-Rad), 150 nM of each primer and ~0.3 ng of cDNA template (except in the case of the housekeeping control, where we used ~0.03 ng). Samples were prepared in duplicate with 10% additional volume. Droplet generation, PCR amplification and droplet analysis were done using the QX200 AutoDG ddPCR system (Bio-Rad). PCR conditions were: 95 °C for 5 min (1×), 95 °C for 1 min and 60.1 °C for 2 min (44×). For all steps a ramp rate of 2 °C/s was used. Data were analysed with the Quantasoft software 1.7.4.0917 (Bio-Rad).

## Statistical analysis
Experiments were performed with at least two biological replicates. To generate data for fixed samples, germaria were collected from at least 5 different adult females grown under equivalent environmental conditions. The average values ± standard error of the mean (SEM) are represented. *P*-values comparing parametric datasets with similar variances were obtained using two-tailed, unpaired Student's *t*-tests. *P*-values of non-parametric samples (Figs. 2b, 3b and 6c) were calculated using the compare proportions test with respect to two conditions. Since the data did not fit a normal distribution, they were analysed in a 2 × 2 table (non-normality test, non-parametric). In all cases, *$P \leq 0.05$, **$P \leq 0.005$, ***$P \leq 0.0005$, ****$P \leq 0.0001$. The distribution of AS events in the four categories (Alt. EX, Alt A., Alt D. and IR; Fig. 2) were calculated using the Chi-square test. *P* values > 0.05 were considered non-significant. CDF analyses were performed using Wilcoxon signed-rank tests. The significance of the KEGG pathway enrichment analyses were verified performing Fisher's exact tests. Violin plots show the first and third quartiles and the median. In all graphs, comparisons without asterisks indicate non-significant differences.

## Reporting summary
Further information on research design is available in the Nature Portfolio Reporting Summary linked to this article.

## Data availability
The datasets generated during the current study are available in the GEO repository, with accession number GSE282708. The process data generated in this study are provided in the Supplementary Information. Source data are provided with this paper.

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

## Acknowledgements

We thank the BDSC, the VDRC and the DSHB (University of Iowa) for fly stocks and antibodies. We thank A. Campoy for help with image quantification, E. Navarrete-Álvarez for support with statistical methods and I. Cases for help with bioinformatic analyses. We thank S. C. Herrera, J. Garrido-Maraver, H. Sánchez-Gómez and M. D. Martín-Bermudo for comments on the manuscript and members of the González-Reyes' laboratory for helpful discussions. The following grants provided financial support: PID2020-115040GB-I00 (MI), PID2021-125480NB-I00 and CEX2020-001088-M (A.G-.R.), funded by MICIU/AEI/10.13039/501100011033; P20_00888 funded by the Junta de Andalucía (A.G-.R.); and by ERDF, EU. D.E-.R. (FPU15/06664) and J.H-R. (FPU17/03230) were funded by FPU contracts from the MECD.

## Author contributions

D.E-.R., J.H-.R., F.Z. and A.G-.R. conceived and designed research; D.E-.R., J.H-.R., G.N. and M.M.M. performed research; D.E-.R., J.H-.R., F.Z., M.B., M.I. and A.G-.R. analysed data; A.G-.R. wrote the paper.

## Competing interests

The authors declare no competing interests.
