## [Peer review file · Nature Communications]

***Drosophila* ovarian stem cell niche ageing involves coordinated changes in transcription and alternative splicing**

Corresponding Author: Professor Acaimo González-Reyes

Version 0:

Reviewer comments:

Reviewer #1

(Remarks to the Author)

Stem cell self-renewal or differentiation is highly regulated by its cellular microenvironment (also known as niche cells) through intercellular signals, direct cell-cell communication, or cellular adhesion. *Drosophila* germline stem cells (GSCs) and their surrounding somatic cells have been used very successfully as a model system to study stem cell regulation in their intact environment. Here, the authors investigated the transcriptomes of two sub-populations of niche cells surrounding female GSCs during aging. At the most anterior part, Terminal Filament cells (TFCs) and Cap cells (CCs) are in direct contact with GSCs with very little turnover; while more posteriorly, Escort cells (ECs) contact GSCs and differentiating germline cells with some turnover due to the presence of some somatic stem cells. The authors isolated TFCs and CCs from ECs using Tau-GFP expressed specifically in these sub-populations and then sorted these two populations by FACS. They then compared the RNA content of these populations in 1-week and 4-week-old flies. In TFCs and CCs, the authors found that many RNAs changed over time, not only in terms of quantity (gene expression) but also in terms of alternative splicing (AS). In contrast, only a few gene expressions changed in ECs both in quantity and alternative splicing. The authors concluded that alternative splicing may play an important role during the aging of TFCs/CCs niche cells and the regulation of GSCs. The authors then focused on the Smu1 gene, a core component of the spliceosome regulating alternative splicing. They found that Smu1 itself undergoes differential splicing with aging in these two populations and is also differently expressed both in level and timing. Using different loss-of-function genetic tools (RNAi and a new specific CRISPR alleles, Smu1), they demonstrated that Smu1 is required in both populations to control the number of GSCs (Figure 5, S4) and the number and morphology of niche cells (Figure 5, S5). They then used RNA sequencing to show that Smu1 controls gene expression and alternative splicing in ECs (Figure 6, S6). This was followed by an analysis of the adhesion protein, Fas2, as Fas2 is known to be a target of Smu1 in larval synapses. They showed that Fas2 is differentially spliced and expressed with aging in TFCs and CCs. They demonstrated that Fas2 is expressed in the niche, with its localization preferentially at the GSC/CC boundary, and that its localization at CC/CC boundaries varies with aging. Finally, they showed that the polarized localization of Fas2 in CCs depends on Smu1 (Figure 7, S7).

I will start by stressing that generating these RNA-seq datasets in wild type and mutant conditions (Smu1) is a great technical achievement, which will be very useful to the community. However, some additional controls could strengthen the authors' conclusions (see second major comment below). The genetic tools used to decipher the functions of Smu1 and Fas2 are also very elegant, and I have no major criticisms at the technical level. My main comment is about how the authors link the dots in their manuscript between the RNA-seq datasets, the loss-of-function of Smu1 and Fas2, and the aging of niche cells. Their finding of cell-type specific ageing signatures at transcription and splicing levels will add, nonetheless to our growing understanding of stem cell niche ageing.

Major comments

1) links between alternative splicing datasets, smu1 gene expression and Fas2 phenotype.

The authors found a number of genes (mainly in TFC/CC) whose expression or splicing changes during aging. Smu1 is one of them in TFC/CC but not in ECs. Since Smu1 is important for splicing, they investigated the consequences of Smu1 loss of function. However, their data show that Smu1 levels increase during aging rather than decrease (Figure 4C). I don't understand why a loss-of-function experiment would then mimic aging. To mimic aging, the authors should overexpress specific isoforms of Smu1 (the ones increasing during aging).

Then, the authors perform RNA-seq on *ptc>Smu1-RNAi* cells, which are mainly EC cells. However, their initial RNA-seq datasets showed that Smu1 expression and splicing changes occurred in TFC/CC cells, not in EC cells. Indeed, they found that (Line309) "Finally, comparing the transcriptome of Smu1-depleted ECs with that of aged ECs revealed that decreasing swiftly Smu1 mRNA levels in ECs did not recapitulate extensively molecular aspects of EC aging." And concluded "We

conclude that the molecular signatures behind EC ageing and the decrease in Smu1 activity within ECs show limited overlap.”

Here, I am not criticizing the loss-of-function analysis of Smu1 on GSC self-renewal and EC morphology, which is very convincing; but I am questioning how this loss-of-function relates to aging.

Similarly, the link between Fas2 lof and aging is not clear. The authors found (line349) “CpC-CpC boundaries contained significantly higher amounts of neural Fas2 in 4w-old samples ($P < 0.0001$; Fig. 7C-D). From these results we conclude that the increase in Fas2 transcription in TFCs-CpCs during ageing is reflected in a surge of Fas2 neural isoforms in CpC-CpC contacts specifically, pointing to an age-dependent polarized distribution of Fas2 isoforms in CpCs”. However, they proceed to analyze a LOF of smu1 (hh>smu1 RNAi) in which Fas2 isoforms are decreased and also directly knock-down Fas2 (hh>Fas2 RNAi) to find that the number of GSCs decreases over time. Again, if the goal is to mimic aging, I would have expected gain of function experiments in which specific isoforms of Fas2 are overexpressed.

Finally, it would be interesting to test if these specific isoforms of Fas2 could rescue Smu1 lof (for example using the MARCM system). It would help connect these two dots (although smu1 lof would still not mimic aging).

2) Purity and contamination of isolated cell populations by FACS

I am confident that the authors were able to efficiently separate TFC/CC cells from EC cells using FACS, as the RNAseq and AS profiles show significant differences. Nonetheless, I am concerned about the potential contamination of both cell populations with other cell types, such as germline cells. While the authors primarily compared the same cell populations at 1 week and 4 weeks, making contamination less of an issue due to the focus on changes within each population, it would still be valuable to assess whether germline cells, for instance, co-purified with TFC or EC cells. It is known that germline cells express numerous neural genes, which might explain the unexpected presence of neural genes in the authors' datasets. (Line 408): “our findings strongly suggest that the ovarian niche cells share some similarities with neural cells.” It would also be interesting to compare these datasets to previously published transcriptomes by bulk sequencing or single-cell RNA sequencing of niche cells (Jevitt et al. 2020; Slaidina et al. 2021; Wilcockson and Ashe 2019; Rust et al. 2020; Samuels et al. 2024). Have these previous studies found similar neural gene expression in niche cells?

Minor comments:

1) Figure 1A : ECs contacting GSC are in yellow, while EC outside of the niche are in brown. This is not mentioned in the legends.

2) Figure 4C: please indicate if the differences are statistically relevant.

3) Line 236 : « First, the amounts of Smu1 mRNA in ECs were 100-400-fold higher than in TFCs-CpCs. » This is not shown.

4) Line 430 : « ECs may be under selective pressure to limit their variability during ageing, as significant transcriptional alterations could potentially have severe implications for the germline ». If so, why the selective pressure would be different for TFCs and ECs ?

5) Line 983 : « Error bars indicate the standard error of the mean (SEM)» . There are no visible error bars in Figure 1E and similar graphs.

6) Line 996 : Figure 2B, Pie charts represent the different AS categories is not indicated.

7) The genetic background is not discussed : Cell adhesion, cell polarity and cytoskeleton GE genes are a large part of altered genes in GE and AS experiment in TFs and CpCs populations. Does overexpression of tau::GFP misregulate such genes ? For example, RT-qPCR of appropriate genes in bab1>tau::GFP and control ovaries could clarify this point.

Reviewer #2

(Remarks to the Author)

Even-Ros et al. present a study examining gene expression and splicing changes in regulating the GSC niche during aging using RNA-seq. This is interesting work. They have identified various genes and splicing events, and characterized the functions of certain genes, such as Fasciclin 2 and Smu1. However, there are some concerns:

Major concerns:

1. The authors observed different alternative splicing (AS) patterns of Smu1 in TFCs-CpCs during aging, but no such regulation was noted in ECs. However, the loss of Smu1 in TFCs-CpCs, ECs, or both can reduce the GSC number, making it challenging to directly link Smu1's function to AS regulation. Furthermore, even in TFCs-CpCs, it remains unclear whether the misregulation of AS of Smu1 is related to aging-induced GSC reduction. The authors should perform the experiments to test it.

2. Expression of Fas2 is upregulated in 4-week-old cells. If this upregulation is related to aging, then the loss of Fas2 function should be able to block the aging phenotype, such as the loss of GSCs. Additionally, overexpression of Fas2 should cause a GSC loss phenotype. However, the loss of Fas2 itself causes GSC loss. Therefore, the gene expression analysis is not consistent with the functional analysis.

Minor concerns:

The authors hypothesize that Smu1 function in CpCs might affect Fas2 splicing. They should perform RT-qPCR to test the expression of different isoforms.

Reviewer #3

(Remarks to the Author)

In their manuscript, "Stem cell niche ageing involves coordinated changes in transcription and alternative splicing," Dilamm and colleagues focused on fly GSC niche aging and compared the transcriptomes of different cell types in young and old flies. They identified shared and unique changes in gene expression and alternative splicing during aging in this context. They further highlighted two regulated genes, Smu1 and Fas2, and performed comprehensive genetic studies, revealing their coordinated role in the aged phenotype through splicing regulation.

Overall, I believe this study deserves publication in Nature Communications. The data will be of interest to researchers in the splicing field, as well as those studying *Drosophila* germline gene regulation and aging. The study provides valuable insights into the intricate mechanisms of aging in fly germline development, highlighting the importance of both GE and AS in this process. The identification of key genes and their roles in germline aging is of great interest and will certainly be a growing topic in the future. I only have a few minor comments on the analysis side.

- Overall, the authors compared GE/AS only with highly regulated genes (cutoff by PSI and fold change), which might underestimate their correlation. They should consider plotting the CDF of gene expression changes of AS genes and vice versa for genome wide comparison.
- Due to the concerns mentioned, the authors should conduct a gene set enrichment analysis (GSEA) in addition to their gene ontology analysis, as GSEA does not require a cutoff. There are several fly GSEA resources available (e.g., PMID: 34710184, PMID: 37125646).
- For quality control purposes, the authors should display marker gene expressions from sorted TFCs-CpCs and ECs.
- The authors focused on two genes and discussed their correlation with ageing and their phenotypes. Although they presented PSI values for each splicing event, it would enhance the clarity of their findings to include actual RNA-seq tracks with sashimi plots of the Smu1 and Fas2 loci.
- As this paper lacks mechanisms, the authors could consider investigating whether alternatively spliced exons have distinctive features, such as exon sizes or sequence preferences in the flanking regions. They could examine spliceosome components and/or known RBP expression changes by aging.
- In line 143, the authors concluded that older samples exhibited a tendency to reduce the inclusion of alternative exons and increase intron retention. However, in Figure 2B, for the Alt EX category, there is a near-equal split of exons being included or excluded in older samples (177 included, 215 excluded). The regulation of A3SS and A5SS events appears more direct. This discrepancy should be clarified.
- The manuscript reports 1,421 differentially expressed genes and 673 alternatively spliced genes (958 events) in aged TFCs-CpCs, but these numbers do not match the venn diagram in Figure 2c. Also, the authors should clarify if there is a direct relationship between the inclusion/exclusion of AS events and increased/decreased gene expression, and whether genes with multiple AS events show stronger GE changes. Highlighting this relationship would strengthen their conclusions.
- In line 315, the authors stated that 53 genes are commonly regulated by AS during ageing and in smu-RNAi experiments, categorizing them as 'same category.' Does this mean that splicing occurred in the same direction on the same exon (as shown in Figure 6, panels g and h)? This should be clarified.
- The Aging Fly Cell Atlas (PMID: 37319212) annotated female germ line cells, including escort cells, from flies aged 5, 30, 50, and 70 days. The authors could cross-reference differentially expressed (DE) genes with this atlas to highlight AS analysis and ensure data quality.

Version 1:

Reviewer comments:

Reviewer #1

(Remarks to the Author)

In this revised version of the manuscript, the authors have addressed all our comments. While the manuscript itself has not undergone significant changes in terms of data or conclusions, several important points have been improved. The authors have thoroughly addressed our criticisms in their rebuttal letter and have included multiple significant new experiments in this revised version.

Our primary criticism was that the manuscript initially seemed somewhat misleading. The authors described changes in gene expression and alternative splicing during the aging of specific cell populations (via RNA-seq) but then conducted mechanistic experiments (loss-of-function or gain-of-function) that did not correspond to the dynamics observed in their RNA-seq data. The authors have now clarified that they are not attempting to replicate aging and explicitly stated that their experiments "did not recapitulate aging." This critical point has been made much clearer.

We still have a few minor remaining points:

3) Line 236 : « First, the amounts of Smu1 mRNA in ECs were 100-400-fold higher than in TFCs-CpCs. » This is not shown. The reviewer is correct. We have modified the text to say 40-150 (the values in the Y axes of Figs. 4E and F are different. Values range from 2,5-3 copies/ng of cDNA in TFCs-CpCs to 100-400 in ECs).

We understand that the authors have already changed the fold changes, but it still does not include the data at 3weeks in TFC/CpC, which is closed to zero (Figure 4d). Why is this data point not taking into account?

7) The genetic background is not discussed: Cell adhesion, cell polarity and cytoskeleton GE genes are a large part of altered genes in GE and AS experiment in TFs and CpCs populations. Does overexpression of tau::GFP misregulate such genes? For example, RT-qPCR of appropriate genes in *bab1>tau::GFP* and control ovaries could clarify this point. This is an important point that we discussed thoroughly while defining the strategy of our approach and that we addressed experimentally. We believe we did the right controls. First, flies expressing tau::GFP with any of the Gal4 drivers utilized were perfectly viable, healthy and fertile and possessed normal-looking ovaries. In fact, we keep stocks containing both the driver and the UAS-tau::GFP construct. Second, we scored GSC numbers and Fas2 localisation in CpC-CpC contacts of *bab1-Gal4*, *hh-Gal4*, *bab1-Gal4 UAS-tau::GFP* and *hh-Gal4 UAS-tau::GFP* 4-week-old flies and could not detect significant differences between them. Thus, tau::GFP expression in niche cells of 4 week-old females does not affect niche activity, at least as measured by GSC numbers and 1D4 levels. Third, our study identified differentially expressed genes in old versus young niche cells but that were treated in the same conditions, i.e. with the same cell sorting protocol and with the same genetic background (expressing tau::GFP). We agree with the reviewer that, should we be studying cells expressing tau::GFP and comparing them with tau::GFP-non expressing cells, we should perform additional controls. In our case, however, the genetic background in both conditions is the same and the expression of the GFP fusion seems to have no deleterious effect on niche activity.

This is a convincing and interesting answer, which should be included in the manuscript. It could be included in the Material and Methods section.

We have added this information to the text and have modified Figs. 7 (panel H) and S7 (panel E).

We could not find panel E on Figure S7.

Several genotypes are missing or wrong for examples:

1) sup Figure S5D : genotypes are missing on line 644 in experimental genotypes

2) sup Figure S9: wrong genotypes, I guess it is *hh>Fas2i* and not *hh>Smu1i*, as indicated in the supplementary figure 9 legends.

Etc...

Reviewer #2

(Remarks to the Author)

My concerns have been addressed.

Reviewer #3

(Remarks to the Author)

In the revised version of their manuscript, the authors have thoroughly addressed all the issues I raised in their previous submission. They have conducted several additional analyses and clarified previously unclear aspects of the study. I am satisfied with their responses and strongly support the publication of this work in Nature Communications.

Detailed response to the reviewers' comments

(We thank the reviewers for their kind words, work and help with the evaluation of this manuscript. Their comments have aided to improve the quality and experimental strength of our work).

Reviewer 1: Stem cell self-renewal or differentiation is highly regulated by its cellular microenvironment (also known as niche cells) through intercellular signals, direct cell-cell communication, or cellular adhesion. *Drosophila* germline stem cells (GSCs) and their surrounding somatic cells have been used very successfully as a model system to study stem cell regulation in their intact environment. Here, the authors investigated the transcriptomes of two sub-populations of niche cells surrounding female GSCs during aging. At the most anterior part, Terminal Filament cells (TFCs) and Cap cells (CCs) are in direct contact with GSCs with very little turnover; while more posteriorly, Escort cells (ECs) contact GSCs and differentiating germline cells with some turnover due to the presence of some somatic stem cells. The authors isolated TFCs and CCs from ECs using Tau-GFP expressed specifically in these sub-populations and then sorted these two populations by FACS. They then compared the RNA content of these populations in 1-week and 4-week-old flies. In TFCs and CCs, the authors found that many RNAs changed over time, not only in terms of quantity (gene expression) but also in terms of alternative splicing (AS). In contrast, only a few gene expressions changed in ECs both in quantity and alternative splicing. The authors concluded that alternative splicing may play an important role during the aging of TFCs/CCs niche cells and the regulation of GSCs. The authors then focused on the *Smu1* gene, a core component of the spliceosome regulating alternative splicing. They found that *Smu1* itself undergoes differential splicing with aging in these two populations and is also differently expressed both in level and timing. Using different loss-of-function genetic tools (RNAi and a new specific CRISPR alleles, *Smu1*), they demonstrated that *Smu1* is required in both populations to control the number of GSCs (Figure 5, S4) and the number and morphology of niche cells (Figure 5, S5). They then used RNA sequencing to show that *Smu1* controls gene expression and alternative splicing in ECs (Figure 6, S6). This was followed by an analysis of the adhesion protein, *Fas2*, as *Fas2* is known to be a target of *Smu1* in larval synapses. They showed that *Fas2* is differentially spliced and expressed with aging in TFCs and CCs. They demonstrated that *Fas2* is expressed in the niche, with its localization preferentially at the GSC/CC boundary, and that its localization at CC/CC boundaries varies with aging. Finally, they showed that the polarized localization of *Fas2* in CCs depends on *Smu1* (Figure 7, S7).

I will start by stressing that generating these RNA-seq datasets in wild type and mutant conditions (*Smu1*) is a great technical achievement, which will be very useful to the community. However, some additional controls could strengthen the authors' conclusions (see second major comment below). The genetic tools used to decipher the functions of *Smu1* and *Fas2* are also very elegant, and I have no major criticisms at the technical level. My main comment is about how the authors link the dots in their manuscript between the RNA-seq datasets, the loss-of-function of *Smu1* and *Fas2*, and the aging of niche cells. Their finding of cell-type specific

ageing signatures at transcription and splicing levels will add, nonetheless to our growing understanding of stem cell niche ageing.

Major comments

R1.1- 1) links between alternative splicing datasets, *smu1* gene expression and Fas2 phenotype. The authors found a number of genes (mainly in TFC/CC) whose expression or splicing changes during aging. *Smu1* is one of them in TFC/CC but not in ECs. Since *Smu1* is important for splicing, they investigated the consequences of *Smu1* loss of function. However, their data show that *Smu1* levels increase during aging rather than decrease (Figure 4C). I don't understand why a loss-of-function experiment would then mimic aging. To mimic aging, the authors should overexpress specific isoforms of *Smu1* (the ones increasing during aging).

We did two different experiments to assess mRNA levels in TFCs-CpCs and ECs, bulk RNA-seq and digital PCR for selected candidates such as *Smu1*. Fig. 4 displays all these data. We have performed statistical analyses of the RNA-seq and ddPCR results and they confirm 1) that there are no significant differences between *Smu1* levels at 1w vs 4w in TFCs-CpCs. There are however between 1w vs 3w and 3w vs 4w, with a clear drop in *Smu1* levels at 3w. 2) There is a slight increase in ECs at 4w that is significantly different to the 1w values in the ddPCR data, but not in the RNA-seq values. Also, the 1w vs 3w and 3w vs 4w are statistically different but, in this case, there is a substantial rise in *Smu1* amounts at 3w. Thus, both cell groups display very different *Smu1* profiles in the monitored 4 weeks.

Considering that *Smu1* is a core component of splicing and that the AS programme of niche support cells is affected by ageing, we wished to manipulate splicing and study in detail the consequences of this manipulation. As the reviewer rightly states, decreasing *Smu1* levels using the RNAi approach does not mimic ageing (well, it would in the case of 3-week old TFCs-CpCs) and this was not our aim. By knocking down *Smu1* mRNAs we wanted to disrupt some aspects of splicing and evaluate the consequences for niche homeostasis.

Following the Reviewer's suggestion and since all *Smu1* isoforms share the same ORF, we have utilised two available lines (BDSC:30061=*Smu1*^{G4103} and BDSC:22072=*Smu1*^{DP01419}) to overexpress *Smu1* in a Gal4/UAS dependent manner in ECs. We used the *ptc-Gal4* driver to determine if we could recapitulate the GSC loss that correlated with an increase in *Smu1* mRNA levels in 3w-old ECs (flies were grown at 18°C and kept at 29°C for 2 weeks upon eclosion). While the 30061 stock did not give significant differences in GSC numbers between control and experimental flies (control: 2.82 GSCs/germ, n=33; *ptc*^{ts}>*Smu1*^{G4103}: 2.48 GSCs/germ, n=31; P=0.12), there was a tendency to lose GSCs in the experimental condition. In addition, the distribution of niches with 1, 2 or 3-4 GSCs reflects the GSC decrease in *ptc*^{ts}> *Smu1*^{G4103} germaria (see chart below). The use of the 22072 line to overexpress *Smu1* did show a significant decrease in the number of GSCs hosted within the niche (control, 2.28 GSCs/germ, n=25; *ptc*^{ts}>*Smu1*^{DP01419}, 1.9 GSCs/germ, n=30, P=0.0408; Fig. S5D). Thus, *Smu1* overexpression in escort cells seems to recapitulate some aspects of niche ageing, at least regarding GSC numbers. The results with the 22072 line have been included in the text and a new panel added to Fig. S5 (S5D).

We scored 33 control and 31 experimental germlines for GSC numbers. Comparisons were made using two-tailed Student's t-test. CONTROL *w; tub-Gal80^{ts}/+; P{w[+mC]=EP}dnk[G4103]/TM6B*. EXPERIMENTAL *w; tub-Gal80^{ts}/+; P{w[+mC]=EP}dnk[G4103]/ptc-Gal4⁴⁵⁹⁰⁰*.

R1.2- Then, the authors perform RNA-seq on *ptc>Smu1-RNAi* cells, which are mainly EC cells. However, their initial RNA-seq datasets showed that *Smu1* expression and splicing changes occurred in TFC/CC cells, not in EC cells. Indeed, they found that (Line309) "Finally, comparing the transcriptome of *Smu1*-depleted ECs with that of aged ECs revealed that decreasing swiftly *Smu1* mRNA levels in ECs did not recapitulate extensively molecular aspects of EC aging." And concluded "We conclude that the molecular signatures behind EC ageing and the decrease in *Smu1* activity within ECs show limited overlap."

Here, I am not criticizing the loss-of-function analysis of *Smu1* on GSC self-renewal and EC morphology, which is very convincing; but I am questioning how this loss-of-function relates to aging.

As stated above, the reason behind analysing the role of *Smu1* in the niche was to assess whether defective splicing influenced niche functioning, not to recapitulate ageing. Our results show that ageing affects AS and that splicing factors other than *Smu1* are also affected by ageing. In fact, we now have some preliminary evidence suggesting a key role of another AS gene (*mb1*) in niche activity. The fact that *Smu1* is important for niche architecture and homeostasis is also reflected in the fact that we found a significant number of GE and AS changes in *Smu1* RNAi ECs. Lastly, since ageing of either TFCs-CpCs or ECs involves differential GE and AS of a large collection of genes, this makes it difficult to recapitulate ageing by manipulating single genes. We have done this in the past with a number of candidates, but the number of "successful" targets is quite limited. We have added a sentence to the *Smu1* part of the Results to emphasize the essential function of *Smu1* regulation in niche homeostasis.

R1.3- Similarly, the link between *Fas2* lof and aging is not clear. The authors found (line349) "CpC-CpC boundaries contained significantly higher amounts of neural *Fas2* in 4w-old samples ($P < 0.0001$; Fig. 7C-D). From these results we conclude that the increase in *Fas2* transcription in TFCs-CpCs during ageing is reflected in a surge of *Fas2* neural isoforms in CpC-CpC contacts specifically, pointing to an age-dependent polarized distribution of *Fas2* isoforms in CpCs". However, they proceed to analyze a LOF of *smu1* (*hh>smu1 RNAi*) in which *Fas2* isoforms are decreased and also directly knock-down *Fas2* (*hh>Fas2 RNAi*) to find that the number of GSCs

decreases over time. Again, if the goal is to mimic aging, I would have expected gain of function experiments in which specific isoforms of Fas2 are overexpressed.

The reviewer is correct. The Fas2 RNAi experiment would not be appropriate to address a putative role for Fas2 in ageing. The idea behind studying Fas2 RNAi was to test if Fas2 was at all needed for niche activity.

In agreement with the reviewer's suggestion, we have checked whether *Fas2* overexpression in young females would recapitulate the GSC loss detected in 4-week-old females and we found that it did not. Rather, we observed an increase in the number of GSCs per niche upon *Fas2* overexpression. We utilized the EP line 1462 (BDSC:11231) to induce *Fas2* overexpression with *hh-Gal4 + tub-Gal80^{ts}* (*hh^{ts}>Fas2^{EP1462}*; flies were grown at 18°C and kept at 29°C for 2 weeks upon eclosion). As a control for the induction of Fas2 ectopic expression due to the EP line, we confirmed the increase in Fas2 levels in experimental *hh^{ts}>Fas2^{EP1462}* CpCs using the 1D4 antibody. Next, we quantified the number of GSCs present in both control and experimental niches and found a significant increase in GSC numbers upon *Fas2* overexpression (control, 2.63 GSCs/germ, n=30; *hh^{ts}>Fas2^{EP1462}*: 3.12 GSCs/germ, n=33; P=0.0047). The interpretation of this result is complex, as with the EP line we most likely induced expression of all *Fas2* isoforms and this may be different to the real scenario in aged cells, in which isoform expression is heterogeneous. Ideally, we would have liked to alter the expression levels and the ratio of the different isoforms (hoping that the intracellular localization of Fas2 isoforms within CpCs would also change as it does in old cells) so that we could reproduce the situation in aged niches, but this is beyond reach.

Considering our previous result showing that *Fas2* knockdown decreased the number of GSCs kept within the niche, the above data reflect a key role for Fas2 levels in maintaining a proper GSC pool and niche architecture. We have added this information to the text and have modified Figs. 7 (panel H) and S7 (panel E).

R1.4- Finally, it would be interesting to test if these specific isoforms of Fas2 could rescue *Smu1* lof (for example using the MARCM system). It would help connect these two dots (although *smu1* lof would still not mimic aging).

We detect a surge in Fas2 expression at 4 weeks (~4.4 fold) and a change in the AS of two of the isoforms, *Fas2-RA* and *-RF* (with a ~5 fold reduction with respect to the 1w levels). We could thus assume that the amounts of *Fas2-RA* and *-RF* in 4w-old cells is about the same as in 1w-old samples — it is the levels of other isoforms that account for the increase in *Fas2* expression in old cells.

The reviewer's suggestion is an interesting one, but not without complications. To try and rescue for instance the GSC loss detected in *hh^{ts}>Smu1 RNAi* niches with the overexpression of *Fas2-RA* and *-RF* would create a novel combination of *Fas2* isoforms in the *hh-Gal4*-expressing cells, significantly complicating the interpretation of the results. In this regard, the fact that overexpressing Fas2 in TFCs-CpCs induces extra GSCs (while one might expect the opposite) illustrates the difficulty in the interpretation of overexpression/rescue experiments.

R1.5- 2) Purity and contamination of isolated cell populations by FACS

I am confident that the authors were able to efficiently separate TFC/CC cells from EC cells using FACS, as the RNAseq and AS profiles show significant differences. Nonetheless, I am concerned about the potential contamination of both cell populations with other cell types, such as germline cells. While the authors primarily compared the same cell populations at 1 week and 4 weeks, making contamination less of an issue due to the focus on changes within each

population, it would still be valuable to assess whether germline cells, for instance, co-purified with TFC or EC cells. It is known that germline cells express numerous neural genes, which might explain the unexpected presence of neural genes in the authors' datasets. (Line 408): "our findings strongly suggest that the ovarian niche cells share some similarities with neural cells." This is a fair comment that we should have addressed in the original submission, as we had the below analyses ready at the time of submission. First, we isolated the cell populations based on GFP expression but also on cell size, to eliminate doublets. Since germline cells are significantly larger than TFCs-CpCs or ECs, we were pretty convinced of the lack of contamination in our samples. Second, we also looked at the germline marker *vasa* in the GE datasets from TFCs-CpCs and ECs and found that it is not present among the 10,000+ genes detected in TFCs-CpCs, the 12,000+ in ECs or the 11,000+ in the *Smu13* RNAi experiment. This has been added to the text. (Please see also our response to comment **R3.3** for extended details of cell population purities).

R1.6- It would also be interesting to compare these datasets to previously published transcriptomes by bulk sequencing or single-cell RNA sequencing of niche cells (Jevitt et al. 2020; Slaidina et al. 2021; Wilcockson and Ashe 2019; Rust et al. 2020; Samuels et al. 2024). Have these previous studies found similar neural gene expression in niche cells? (Unless we are missing something, Wilcockson + Ashe 2019 and Samuels et al. 2024 only did RNA-seq of germline cells).

Thanks for the useful suggestion. In fact, the works of Jevitt et al. and Rust et al. report expression of nervous system development/related to neural activity genes in GSC niche cells. We have analysed the list of TFC, CpC and EC genes in Slaidina et al. and found that some of them also group in categories related to neural activity such as axon growth, synaptic target transmission or neural development (see below for some GO_BP examples). These data have been added to the text.

A	B	C	D	E	F	G	H	I	J	K	L	M
Category	Term	Count	%	Enrichment	Genes	List Total	Pop Hits	Pop Total	Fold Enrichment	Benferroni	Benjamini	FDR
1	GOTERM_BP_DIRECT GO:0007411 axon guidance	25	5	1.53E-08	NETA, ROBO2, SEMA5A, D, OTX, BDL, INH, NRY, SBB, SDC, FAS1, FAS2, SM, TUTL, UNC-115A, BETA	460	177	12975	4	2.16E-05	2.16E-05	2.10E-05
2	GOTERM_BP_DIRECT GO:0048511 animal organ development	11	2	1.14E-06	BR, DRPL, AB, TENA, LANBL, OTX, VN, EN, STY, INK, TRO1	460	41	12975	8	0.00165538	6.83E-04	6.62E-04
3	GOTERM_BP_DIRECT GO:0007135 cell adhesion	13	3	1.44E-06	GLEC, MMP1, CG1739, OTX, IRW, HOW, TENA, FER, NIA, FAS1, FAS2, CAPS, TIMP	460	62	12975	6	0.002047629	6.83E-04	6.62E-04
4	GOTERM_BP_DIRECT GO:0007150 homophilic cell adhesion via plasma membrane adhesion molecules	12	2	2.82E-06	ROBO2, PLAK, CDN, FZ, DTK, CADN, FAS1, FAS2, BDL, TRO1, CAPS, TUTL	460	55	12975	6	0.004008065	0.00104028	9.72E-04
5	GOTERM_BP_DIRECT GO:0000205 positive regulation of neurite proliferation	9	2	3.64E-06	HL, ZLD, ESRP1, MBTA-HL, KLU, ESRP1, SHL, BLN, INK, TRO1, HSP83	460	28	12975	9	0.005456208	0.001094227	0.00109649
6	GOTERM_BP_DIRECT GO:0042026 protein refolding	9	2	1.15E-05	HSC70-4, HSP26, HSP67C, CG14207, HSP27, HSP83, HSP23, HSP70BC, ROE1	460	32	12975	8	0.016212544	0.002571253	0.002489999
7	GOTERM_BP_DIRECT GO:0016059 Wnt signaling pathway	10	2	1.26E-05	SH, WNT6, FJ, NOD, TSH, NOTUM, DIAP1, WNT4, FZ2, WLS	460	42	12975	7	0.017837674	0.002571253	0.002489999
8	GOTERM_BP_DIRECT GO:0016477 cell migration	12	2	1.78E-05	HOW, FHL, HSP90, SOC, RHOGAP2, JARID1, DAD, TRO1, CAPS, TKV, WNT4, FZ2	460	66	12975	5	0.025646464	0.002869318	0.002819156
9	GOTERM_BP_DIRECT GO:0067479 glutathione metabolic process	10	2	1.89E-05	GSTE14, GCLC, GSTD9, GSTT4, GSTD3, GSTS1, GSTE2, GSTD3, GSTE1, GSTE11	460	44	12975	6	0.02651564	0.002869318	0.002819156
10	GOTERM_BP_DIRECT GO:0007476 imaginal disc-derived wing morphogenesis	17	3	2.44E-05	HL, LANBL, DPP, DAD, CV-2, TSH, GCLC, SBB, BN, VN, MAM, MIR, EMC, PTC, TKV, FNG, WLS	460	136	12975	4	0.03418424	0.00478216	0.0033683
11	GOTERM_BP_DIRECT GO:0007624 open tracheal system development	14	3	2.84E-05	NRY2, MMP1, MMP3, VARI, INH, PEG, PTPD, CDK4, CV-C, VIL, RAC2, STY, PTD, TKV	460	95	12975	4	0.036353393	0.00367518	0.003560336
12	GOTERM_BP_DIRECT GO:0008045 motor neuron axon guidance	12	2	4.14E-05	NETA, PTPD, TENA, SDC, SEMA1A, RAC2, FAS2, TRO1, CHER, CAPS, WNT4, FZ2	460	72	12975	5	0.057319372	0.004917388	0.004761944
13	GOTERM_BP_DIRECT GO:0016043 cellular component organization	13	3	5.31E-05	NETA, AB, KEX1, CDN, GEFMESO, DRPR, BR, IRW, PURA, FER, LKB, CADN, TRO1	460	87	12975	4	0.072774945	0.005689511	0.005509716
14	GOTERM_BP_DIRECT GO:0008840 determination of adult lifespan	19	4	5.59E-05	MAGU, ATGBA, SESB, MIP12, MIR, CHER, THOR, ATPALPHA, NIAZ, HSP26, SAMA-5, MEN, HSP27, C	460	177	12975	3	0.07656576	0.005689511	0.005509716
15	GOTERM_BP_DIRECT GO:0035155 regulation of tube length, open tracheal system	8	2	6.65E-05	ATPALPHA, NRY2, FJ, FZ, MMP1, SAND, FAS2, VARI	460	30	12975	8	0.090307706	0.006309715	0.006110521
16	GOTERM_BP_DIRECT GO:0000122 negative regulation of transcription by RNA polymerase II	22	4	8.89E-05	AB, PSC-D, TSH, CVO, EN, GBA, DSP1, BR, EIP75B, MNT, SBB, COOP, CG1620, CG9932, ESRP1, MBT	460	234	12975	3	0.117821132	0.007834883	0.007587098
17	GOTERM_BP_DIRECT GO:0008039 synaptic target recognition	8	2	1.89E-04	SH, ROBO2, NP2A, AB, TENA, SEMA1A, GLL, TUTL	460	35	12975	6	0.235316662	0.015780436	0.015281756
18	GOTERM_BP_DIRECT GO:0035014 cell differentiation	16	3	2.38E-04	MW, D, DAD, ESRP1, MBT-HL, F5, EIP75B, HOW, ESRP1, MBTA-HL, HNF4, CADN, FAS2, EMC, SLS	460	148	12975	5	0.287877166	0.017902182	0.017336453
19	GOTERM_BP_DIRECT GO:0007163 establishment or maintenance of cell polarity	7	1	2.39E-04	KAY, FZ, RHOL, CADN, RAC2, FZ3, ROBO1B	460	26	12975	8	0.28835927	0.017902182	0.017336453
20	GOTERM_BP_DIRECT GO:0007498 mesoderm development	10	2	3.07E-04	HOW, CG4445, HSP90, SIK4, RHOL, CG4447, DPP, SLS, TSH, NIMB2	460	62	12975	5	0.34465226	0.021880447	0.0211189
21	GOTERM_BP_DIRECT GO:0074859 photoreceptor cell axon guidance	7	1	4.50E-04	UNC-115A, SEMA1A, OTX, BDL, FAS1, PEB, CAPS	460	29	12975	7	0.475219168	0.02619591	0.026683578
22	GOTERM_BP_DIRECT GO:0006353 regulation of transcription by RNA polymerase II	41	8	4.58E-04	KAY, D, REPTOR, TMA, DL, CVO, GILUF4E, BR, DSP1, COOP, MNT, SBB, ZLD, SIK4, HNF4, KLU, MYC	460	650	12975	2	0.47882711	0.02619591	0.026683578
23	GOTERM_BP_DIRECT GO:0031290 retinal ganglion cell axon guidance	5	0.96717988	6.10E-04	DAD, OTX, CADN, WNT4, FZ2	460	12	12975	12	0.58688287	0.037777196	0.036583394
24	GOTERM_BP_DIRECT GO:0060070 canonical Wnt signaling pathway	6	1	6.89E-04	WNT6, CJE, FZ, FZ3, WNT4, FZ2	460	21	12975	8	0.62466837	0.032926454	0.032803495
25	GOTERM_BP_DIRECT GO:0061077 chaperone-mediated protein folding	6	1	6.89E-04	HSC70-4, HSP26, HSP67C, CG14207, HSP27, HSP23	460	21	12975	8	0.62466837	0.032926454	0.032803495
26	GOTERM_BP_DIRECT GO:0046844 dorsal appendage formation	9	2	7.37E-04	BR, ARM, DPP, MIR, RAC2, BLN, EMC, SYP, PUC	460	56	12975	5	0.649987836	0.040361558	0.039086087
27	GOTERM_BP_DIRECT GO:0008253 head involution	8	2	9.37E-04	SCV, CV-C, TSH, RAC2, CHB, PTD, EMC, REB	460	45	12975	5	0.75023807	0.048432085	0.047817044
28	GOTERM_BP_DIRECT GO:0070593 dendrite self-avoidance	6	1	0.00107576	ROBO2, PLAK, OTX, FAS2, BDL, TUTL	460	23	12975	7	0.78931276	0.054660545	0.052952262
29	GOTERM_BP_DIRECT GO:0045927 positive regulation of growth	5	0.96717988	0.001166278	MW, NRP2-B, MYC, WAP, TKV	460	14	12975	10	0.810193928	0.057288277	0.054548335
30	GOTERM_BP_DIRECT GO:0051085 chaperone cofactor-dependent protein refolding	7	1	0.00126298	HSC70-4, DNA-1, CSD101, DPP, HSP98, CG11217, HSP70BC	460	35	12975	6	0.84045656	0.0510526	0.050110813
31	GOTERM_BP_DIRECT GO:0035556 moon-canonical Wnt signaling pathway	4	0.773694991	0.001384044	FZ, FZ3, WNT4, FZ2	460	7	12975	16	0.860855746	0.06357674	0.061567643
32	GOTERM_BP_DIRECT GO:0051124 synaptic assembly at neuromuscular junction	7	1	0.001498192	TENA, AY, ATPALPHA, SPIN, SESB, PRODAP, STAI	460	36	12975	5	0.881758395	0.066669547	0.064562715
33	GOTERM_BP_DIRECT GO:0007616 long-term memory	10	2	0.001678901	P51, 18W, PTPD, ARM, MORF, FAPB, MGL, GCLM, CHER, GBA	460	78	12975	4	0.908622448	0.072447122	0.070157711
34	GOTERM_BP_DIRECT GO:0048713 system development	6	1	0.001926365	DRPR, NETA, CADN, DL, TRO1, TUTL	460	26	12975	7	0.935802391	0.080660999	0.078131098
35	GOTERM_BP_DIRECT GO:0007413 axonal fasciculation	5	0.96717988	0.002004591	NRY, CON, HSC70-4, CADN, FAS2	460	16	12975	9	0.942582296	0.081558219	0.078980888

Minor comments:

- Figure 1A : ECs contacting GSC are in yellow, while EC outside of the niche are in brown. This is not mentioned in the legends. This has been corrected.
- Figure 4C: please indicate if the differences are statistically relevant.

They are not. Throughout the paper, we only indicated those differences that are statistically significant. This is mentioned in M+M. In Figs. 4 and 7 we used two-tailed, unpaired with equal variance Student's t-tests, as stated in the respective figure legends.

3) Line 236 : « First, the amounts of Smu1 mRNA in ECs were 100-400-fold higher than in TFCs-CpCs. » This is not shown.

The reviewer is correct. We have modified the text to say 40-150 (the values in the Y axes of Figs. 4E and F are different. Values range from ~2,5-3 copies/ng of cDNA in TFCs-CpCs to ~100-400 in ECs).

4) Line 430 : « ECs may be under selective pressure to limit their variability during ageing, as significant transcriptional alterations could potentially have severe implications for the germline ». If so, why the selective pressure would be different for TFCs and ECs ?

Fair point. While our results prove that distinct cell types within the same organ can age differently, the reasons behind this difference are uncertain. We postulate that the functions ECs play during oogenesis require more stable GE programmes than TFCs-CpCs and propose that ECs are under more stringent selection conditions than TFCs-CpCs (simply put, if the female is to assign resources for egg chamber development, better do it to cysts that have differentiated properly and that can give rise to functional gametes; hence, losing GSCs due to ageing may not be as detrimental as using valuable resources to grow a doomed egg chamber). We have clarified our message stating now that germline cells remain in close contact with ECs during early oogenesis, and it is known that the different EC populations are required for cyst differentiation. Thus, alterations in this physical relationship may affect germ line development. Conversely, the physical CpC-GSC interaction is less dynamic and relies on *DE-Cad*-mediated adhesion mainly. Alternatively, since ECs undergo —albeit limited— cell replenishment during adulthood (see Reilin 2017, NCB), perhaps ECs cells in 4w-old females are not as old as TFCs-CpCs from the same females (TFCs-CpCs do not divide in the adult) and possess more “stable” transcriptomes.

5) Line 983 : « Error bars indicate the standard error of the mean (SEM)» . There are no visible error bars in Figure 1E and similar graphs.

Error bars are represented in Fig. 1B, not in Fig. 1E. We have added a note to the fig. legend to clarify this.

6) Line 996 : Figure 2B, Pie charts represent the different AS categories is not indicated.

Thanks. We have added a sentence explaining the meaning of the pie charts.

7) The genetic background is not discussed: Cell adhesion, cell polarity and cytoskeleton GE genes are a large part of altered genes in GE and AS experiment in TFs and CpCs populations. Does overexpression of *tau::GFP* misregulate such genes? For example, RT-qPCR of appropriate genes in *bab1>tau::GFP* and control ovaries could clarify this point..

This is an important point that we discussed thoroughly while defining the strategy of our approach and that we addressed experimentally. We believe we did the right controls. First, flies expressing *tau::GFP* with any of the *Gal4* drivers utilized were perfectly viable, healthy and fertile and possessed normal-looking ovaries. In fact, we keep stocks containing both the driver and the *UAS-tau::GFP* construct. Second, we scored GSC numbers and Fas2 localisation in CpC-CpC contacts of *bab1-Gal4*, *hh-Gal4*, *bab1-Gal4 UAS-tau::GFP* and *hh-Gal4 UAS-tau::GFP* 4-

week-old flies and could not detect significant differences between them. Thus, *tau::GFP* expression in niche cells of 4 week-old females does not affect niche activity, at least as measured by GSC numbers and 1D4 levels. Third, our study identified differentially expressed genes in old *versus* young niche cells but that were treated in the same conditions, i.e. with the same cell sorting protocol and with the same genetic background (expressing *tau::GFP*). We agree with the reviewer that, should we be studying cells expressing *tau::GFP* and comparing them with *tau::GFP*-non expressing cells, we should perform additional controls. In our case, however, the genetic background in both conditions is the same and the expression of the GFP fusion seems to have no deleterious effect on niche activity.

Reviewer 2: Even-Ros et al. present a study examining gene expression and splicing changes in regulating the GSC niche during aging using RNA-seq. This is interesting work. They have identified various genes and splicing events, and characterized the functions of certain genes, such as Fasciclin 2 and Smu1. However, there are some concerns:

Major concerns:

R2.1- The authors observed different alternative splicing (AS) patterns of Smu1 in TFCs-CpCs during aging, but no such regulation was noted in ECs. However, the loss of Smu1 in TFCs-CpCs, ECs, or both can reduce the GSC number, making it challenging to directly link Smu1's function to AS regulation. Furthermore, even in TFCs-CpCs, it remains unclear whether the misregulation of AS of Smu1 is related to aging-induced GSC reduction. The authors should perform the experiments to test it.

This is an important point. Our transcriptomic analyses of *Smu1* RNAi in ECs demonstrate that *Smu1* knockdown influences AS. Hence, *Smu1* activity in the GSC niche regulates AS. We also describe how ageing affects *Smu1* AS, as nearly 20% of the transcripts (*Smu1* gives rise to 3 different isoforms, but they all share the same coding region) of 4w-old TFCs-CpCs incorporate an intron that introduces an early stop codon, most likely reducing *Smu1* expression by 20% due to non-sense mediated decay. Conversely, ECs do not show AS changes in *Smu1* with age. From this and other results we conclude that different niche cells age with distinctive dynamics. Our results do not allow us to claim that “*the misregulation of AS of Smu1 is related to aging-induced GSC reduction*”. It is true that aged niches contain significantly fewer GSCs than younger ones, and that decreasing *Smu1* levels in CpCs or ECs causes a similar phenotype. However, GSC populations are controlled by a large number of factors and the fact that both conditions (ageing and *Smu1* RNAi) show a similar effect on GSC numbers does not imply that they are both acting on related pathways.

Following the reviewer's comment and in order to clarify the connection between *Smu1*'s dynamic transcription during ageing and GSC loss, we have overexpressed *Smu1* in ECs utilising the *ptc-Gal4* driver in an attempt to recapitulate the profile of *Smu1* expression in ageing ECs. Please see our response to point **R1.1**.

R2.2- Expression of Fas2 is upregulated in 4-week-old cells. If this upregulation is related to aging, then the loss of Fas2 function should be able to block the aging phenotype, such as the loss of GSCs. Additionally, overexpression of Fas2 should cause a GSC loss phenotype. However, the loss of Fas2 itself causes GSC loss. Therefore, the gene expression analysis is not consistent with the functional analysis.

The reviewer is right. The result of GSC loss in *Fas2* RNAi niches was somehow unexpected, as we could have anticipated a (partial) rescue of the age-related GSC loss considering that *Fas2* is

upregulated in old TFCs-CpCs. The problem likely stems from the role of Fas2 in cell adhesion, as we have observed that Fas2 depletion in the *hh>Fas2 RNAi* condition can disturb the integrity of the CpC rosette. In fact, the increase in Fas2 levels during ageing may be a response to impaired cell adhesion in old niche cells. Alternatively or in addition, removing a single gene in the niche cells does not always suffice to produce a GSC phenotype, as we have often experienced before with other candidate genes.

In any case, we have decided to extend our phenotypic analysis and to study the effect of Fas2 overexpression during ageing. Please see our response to Reviewer #1's concern **R1.3** regarding Fas2 overexpression.

Minor concerns:

The authors hypothesize that Smu1 function in CpCs might affect Fas2 splicing. They should perform RT-qPCR to test the expression of different isoforms.

We assume that the reviewer is referring to testing Fas2 isoform composition in TFCs-CpCs in a Smu1 RNAi background. We agree with the reviewer that obtaining these data would be interesting, but we encountered a technical issue here. We have tried hard to find an experimental condition in which we could i) overexpress the Smu1 RNAi construct in TFCs-CpCs while at the same time ii) label these cell types so that we could cell-sort them and perform the transcriptomic analyses. To achieve this, we first tried the *hh-Gal4* driver to induce both UAS-Smu1 RNAi and UAS-*tau::GFP* but to no avail – we could not recapitulate the GSC loss phenotype that we consistently observe in *hh-Gal4 UAS-Smu1 RNAi* females, most likely due to the titration effect of the presence of 2x UAS constructs in the *hh^{ts}>Smu1 RNAi + tau::GFP* flies. Next, we searched for a *lexA* line that we could use to label the TFCs-CpCs to aid us in the cell sorting of this populations of cells, much as we did with the *ptc-lexA* line for the ECs. We could not find any among the available *hh-lexA* stocks. Third, we then looked for a GFP trap expressed strongly in these cell types so that it could be used in the cell sorter, but without success, either because they were not TFC+CpC-specific or because the signal was not strong enough as to allow cell sorting. Fourth, we made a GFP-expressing construct in which GFP expression was controlled by a *hh* enhancer that had been reported to direct gene expression in TFCs and CpCs (GMR28E11). Unfortunately, we could not get transgenic flies after trying 3 times to obtain the transformants. In summary, we were unable to manipulate Smu1 in TFCs-CpCs and, at the same time, isolate them to extract their mRNA and perform RT-qPCR and the transcriptomic analyses.

Reviewer 3: In their manuscript, “Stem cell niche ageing involves coordinated changes in transcription and alternative splicing,” Dilamm and colleagues focused on fly GSC niche aging and compared the transcriptomes of different cell types in young and old flies. They identified shared and unique changes in gene expression and alternative splicing during aging in this context. They further highlighted two regulated genes, Smu1 and Fas2, and performed comprehensive genetic studies, revealing their coordinated role in the aged phenotype through splicing regulation.

Overall, I believe this study deserves publication in Nature Communications. The data will be of interest to researchers in the splicing field, as well as those studying Drosophila germline gene regulation and aging. The study provides valuable insights into the intricate mechanisms of aging in fly germline development, highlighting the importance of both GE and AS in this process. The identification of key genes and their roles in germline aging is of great interest and

will certainly be a growing topic in the future. I only have a few minor comments on the analysis side.

R3.1- Overall, the authors compared GE/AS only with highly regulated genes (cutoff by PSI and fold change), which might underestimate their correlation. They should consider plotting the CDF of gene expression changes of AS genes and vice versa for genome wide comparison.

We have generated a CDF (Cumulative Distribution Function) of gene expression (GE) changes specifically for genes with differential alternative splicing (DAS) events. The reverse analysis would produce results complicated to analyse, as a gene may exhibit changes in GE while being associated with multiple AS events, making it challenging to establish a clear relationship between the two processes.

Attached are the graphs for the three cell types, along with their respective p-values. Wilcoxon tests were conducted using different groups of control genes (sets of genes with no DAS and of the same size as the DAS gene set). The results indicate that the only cell type consistently showing statistically significant differences in gene expression between DAS genes and non-DAS genes is TFCs-CpCs. Specifically, DAS genes tend to be more downregulated at the gene expression level. This information has been added to the text and to Suppl. Figs. 2, 3 and 7.

TFCs-CpCs - p-value = 5.112e-07

ECs - p-value = 0.4767

Smu1_iRNA - p-value = 0.1566

R3.2- Due to the concerns mentioned, the authors should conduct a gene set enrichment analysis (GSEA) in addition to their gene ontology analysis, as GSEA does not require a cutoff. There are several fly GSEA resources available (e.g., PMID: 34710184, PMID: 37125646).

We should have made clear in the original submission that the number of expressed genes in any of the experimental conditions is very high. We identified 10,491 genes expressed in TFCs-CpCs and 12,172 in ECs (1w versus 4w), and 11,783 in ECs (control versus experimental *Smu1* RNAi). Considering that the *D. melanogaster* genome contains 17,728 annotated genes (including protein coding, rRNAs, tRNAs, miRNAs, LncRNAs, etc.; Kaufman, 2017, *Genetics* **206**: 665-689), our RNA seq datasets identified between 60 and 70% of all *Drosophila* genes. Thus, conducting enrichment analyses on all identified genes without setting thresholds to filter the candidates in the specific conditions (1w versus 4w or control versus RNAi) was not prioritised in our studies.

Nevertheless, we have followed the reviewer's suggestion and conducted GSEA analyses utilising PANGEA, one of the platforms mentioned by the reviewer, on the three sets of experimental data reported in our study: 1w-4w ageing in TFCs-CpCs and ECs, and *Smu1* knockdown in ECs. First, we defined the enriched terms (GO_BP, GO_CC and GO_MF) and pathways (KEGG) of all genes identified in TFCs-CpCs at 1w and 4w, ECs at 1w and 4w and *Smu1* RNAi control and experimental samples. As we did in the original submission, we applied a p-value cut-off of $P < 0.01$ for GO terms analyses and of < 0.1 for KEGG pathways. Next, in order to recognise terms that changed with ageing or upon *Smu1* knockdown, we distinguished those unique terms present in both conditions. As seen in the new datasets (Pangea analyses_ALL) we identified a larger number of GO terms in the TFCs-CpCs experiment than in the others. This recapitulates our original findings with the filtered datasets, in which the list of differentially expressed genes in TFCs-CpCs is significantly larger (1421) than in ECs (122) or *Smu1* RNAi (118). In our PANGEA analyses we did not pinpoint obvious, new enriched categories of GO terms beyond those identified using the data filtered for fold change, dPSI and p-value. We have incorporated the new datasets to the revised version and added an explanatory paragraph to the text.

R3.3- For quality control purposes, the authors should display marker gene expressions from sorted TFCs-CpCs and ECs.

The reviewer is right. Please see our response to comment **R1.5** regarding the absence of contamination with germline cells. As for TFC-CpC markers, we have looked at *Lam-C*, which is highly expressed in these cell types and much lower in ECs, and *engrailed*, expressed in TFCs-CpCs but not in ECs. We find that, in our datasets, *Lam-C* levels are 17x (1 week) or 33x (4 weeks) higher in TFCs-CpCs than in ECs, while *engrailed* levels are 22x (1 week) or 102x (4 weeks) higher in TFCs-CpCs. As for EC markers, we have compared the top 50 genes expressed in each of the three EC types (anterior, central and posterior) reported in Rust et al. (Nat. Comms., 2020) with our TFCs-CpCs and ECs datasets. We have identified 3 genes (CG17321, CG6067 and SPE) expressed in both the Rust et al. and our EC lists but not in our TFCs-CpCs list. From all these analyses we conclude that there is no significant contamination of ECs in the TFC-CpC experiment and vice versa. These data have been added to the text.

R3.4- The authors focused on two genes and discussed their correlation with ageing and their phenotypes. Although they presented PSI values for each splicing event, it would enhance the clarity of their findings to include actual RNA-seq tracks with sashimi plots of the *Smu1* and *Fas2* loci.

Thank you for the suggestion. We have added two new supplementary figures (Suppl. Fig. 4 and 8) containing the sashimi plots for the *Smu1* IR event and the *Fas2* exons AS.

R3.5- As this paper lacks mechanisms, the authors could consider investigating whether alternatively spliced exons have distinctive features, such as exon sizes or sequence preferences in the flanking regions.

We have used the Matt toolkit (Gohr and Irimia, Bioinformatics 2019, 35: 130-132) to analyse features of AS exons. To follow are plots of the three datasets (TFCs-CpCs, ECs and *Smu1* RNAi) showing the distribution of different features for exons UP regulated or DOWN regulated. We used three "control" or reference sets for our analyses: AS_NC (alternative axons that do not change in the comparison), CR (cryptic exons; i.e. very lowly included, usually related to splicing errors) and CS (constitutive exons). All statistics are done comparing with the CS reference. In

summary, what our analyses indicate is that there are no major differences between UP and DOWN-regulated exons and that these look roughly similar to other alternative exons. We can also provide the reviewer with the complete “raw” data generated by the Mat tool (these are large pdf files, one for each of the cell types/experiments). Considering that the manuscript is already above the word limit of the journal, we prefer not to include this (negative) result in the text.

TFCs-CpCs

ECs

Smu1 RNAi ECs

They could examine spliceosome components and/or known RBP expression changes by aging. Following the reviewer’s suggestion, we have analysed in detail the expression of genes belonging to the “spliceosomal complexes” (SPL-C) group from the FlyBase Gene groups section in our datasets. We have found that, of the 146 genes belonging to the SPL-C group, 119 are expressed in 1w- and 4w-old TFCs-CpCs, in ECs and in *Smu1 RNAi* ECs. Of those 119, 11 (TFCs-CpCs), 4 (ECs) and 1 (*Smu1 RNAi*) have a fold change <0.5 or >2 when comparing 1w versus 4w or controls versus the *Smu1 RNAi* condition. Finally, when applying the FDR<10% filter, only 4 genes in the TFCs-CpCs samples were identified (*Hsc70-1*, *cdc2rk*, *Es2* and *CG11964*) and only 1 (*Smu1*; not surprisingly, since here we were targeting precisely *Smu1*) in the *Smu1 RNAi* set. From these we conclude that the expression levels of spliceosome components do not seem to be affected specifically during ageing or upon *Smu1* knock-down in niche cells.

R3.6- In line 143, the authors concluded that older samples exhibited a tendency to reduce the inclusion of alternative exons and increase intron retention. However, in Figure 2B, for the Alt EX category, there is a near-equal split of exons being included or excluded in older samples (177 included, 215 excluded). The regulation of A3SS and A5SS events appears more direct. This discrepancy should be clarified.

Regarding the Alt. EX category. As stated in the M+M section, the statistical analyses of the AS events (Fig. 2B, 3B and 6C) were performed using the “compare proportions test with respect to two conditions”. Since the data did not fit a normal distribution, they were analysed in a 2x2 table (non-normality test, non-parametric). This test is a statistical method used to determine if there is a significant difference between the proportions of two groups (in our case, + or - dPSI) based on categorical data. While the reviewer may perceive the distribution of positive

versus negative EX events in Fig. 2B to be similar to that in 3B, the proportion of Alt EX events in the TFCs-CpCs dataset is very different to that of ECs (see the pie charts, white sections). This may explain why the difference in Alt EX distribution in Fig. 2B is statistically significant, while the one in 3B is not.

With respect to A3SS and A5SS. In these categories, only positive values are considered (i.e. alternative splice sites which increase their usage). Since the total usage of all competing splice sites in a given A3SS or A5SS event must be equal to 100, for each alternative donor (or acceptor) that is upregulated there must be other(s) that are downregulated. For this reason, vast-tools counts only the splice sites with "increased splice site usage", thus the positive values.

R3.7- The manuscript reports 1,421 differentially expressed genes and 673 alternatively spliced genes (958 events) in aged TFCs-CpCs, but these numbers do not match the venn diagram in Figure 2c.

The reviewer is right — we made a mistake when writing the numbers in the Venn diagram. Our apologies. This has been corrected (249+709=958 events).

Also, the authors should clarify if there is a direct relationship between the inclusion/exclusion of AS events and increased/decreased gene expression, and whether genes with multiple AS events show stronger GE changes. Highlighting this relationship would strengthen their conclusions.

We have analyzed this relationship with two approaches:

1.- Comparing the log2FoldChange of genes with a single AS event versus those with multiple events. Only for the TFCs-CpCs samples genes with >1 DAS tend to have a subtle, significant downregulation. In the rest of samples, we found no statistically significant differences. To follow are the graphical representations of the data regarding genes with 1 DAS versus >1 DAS in all three samples.

TFCs-CpCs - p-value = 0.005109

ECs - p-value = 0.3506

Smu1_iRNA: p-value = 0.2059

2.- A chi-squared test was performed to assess whether the intersection between genes with changes in AS and GE is significant. Only genes with sufficient read coverage in both AS and GE analyses were considered, which is why the numbers differ from those reported in the paper for each analysis (GE or AS) separately. The conclusion from this study is that, only in the case of the TFCs-CpCs samples, there is a direct, significant relationship between differential GE and AS. Thus, if a TFCs-CpCs gene changes its expression levels, it has a higher probability of also undergoing DAS than if it does not have differential GE. As this is best viewed in the CDF plots - already added to the text and figures-, we have opted not to include this result in the text.

TFCs-CpCs: X-squared = 73.206, df = 1, p-value < 2.2e-16

	AS_yes	AS_no
GE_yes	172	695
GE_no	497	4564

In the other cell types, this relationship is not significant.

R3.8- In line 315, the authors stated that 53 genes are commonly regulated by AS during ageing and in *smu*-RNAi experiments, categorizing them as 'same category.' Does this mean that splicing occurred in the same direction on the same exon (as shown in Figure 6, panels g and h)? This should be clarified.

In our analysis of *Smu1* RNAi ECs, we found 210 AS events corresponding to 189 genes. When compared to the list of genes with AS events during EC ageing, we detected 53 genes in common. Of those, 35 genes (65 events) had events in the same category (Alt EX, Alt A, Alt D or IR); the other 18 genes showed events in different categories in EC ageing and in *Smu1* RNAi. Of the 65 events (35 genes), only 12 events (12 genes) were identical events between EC ageing and *Smu1* RNAi, but only 8 events (8 genes) had the same kind of regulation, either up or down. Thus, only 8 events out of 210 were similar between both groups. We have modified the text and Fig. 6H to make this clearer.

R3.9- The Aging Fly Cell Atlas (PMID: 37319212) annotated female germ line cells, including escort cells, from flies aged 5, 30, 50, and 70 days. The authors could cross-reference differentially expressed (DE) genes with this atlas to highlight AS analysis and ensure data quality.

Thank you for the suggestion. We have retrieved the list of genes expressed in the cell group named "escort cells" that change their expression between 5 and 30 days (5d-30d; we assume that the authors classified, like us, the three types of ECs under one group only) reported in the Lu et al. publication mentioned by the reviewer. After filtering their data for FDR<10% and Fold change>2 (the same as our data), we obtained a list of 233 genes differentially expressed in 30d-old ECs compared to 5d samples. When cross-referenced to our 122-gene list (1w versus 4w escort cells), we only found 1 common gene, *Pde9*. Next, we checked the list of EC genes showing AS changes during ageing (1w-4w; 639 genes) against the 5d-30d dataset and identified 19 genes in common.

The reasons for this minute overlap are not clear to us:

1- Our sequencing depth is most likely much larger than the Ageing Atlas, as theirs was a single-cell approach and we needed at least 40+ RPKMs to detect exon-exon junctions and to assess AS events. This may explain why the GO analysis of the Ageing Atlas 5d-30d data gave a large number of terms related to ribosomes, transcription, translation, nucleolus, etc. In fact, of the 179/233 genes listed in the GO term output (54/233 were not included in the output), nearly 50% (86) of them were related to ribosomal function or nucleolus, with highly transcribed genes. Thus, the Ageing Atlas data may be biased towards most abundant genes.

2- The conditions for sample isolation were different. We used a *y w* stock that we have kept in our lab for a long time. The Ageing Atlas experiments were performed with a

different stock (*w* crossed to OREGON R), fed different food and most likely with different microbiota.

3- It may well be that ageing does not affect simply gene expression but the regulation of gene groups. For instance, it could be that ageing dysregulates biological processes rather than specific genes or that ageing disturbs larger 3D chromatin domains such as TADs (topologically associated domains). In this scenario, the particular genes that are affected by ageing may not be the only indication of common ageing effects between different conditions but also the process in which the genes are involved or the physical location of the changed genes (but we have mapped the 122 1w-4w and the 233 5d-30d genes onto the physical map of the *Drosophila* genome and we cannot define significant gene clusters containing genes from both lists).

4- The assignment of “escort cell fate” to the identified cells in the single-cell experiment may require some attention. We have checked that our sorted cells express ECs markers such as *ptc*, *fax*, *sns* and *hbs* at detectable levels (average expression at 1w: 4475, 34159, 121 and 49; average expression at 4w: 4777, 31602, 98 and 55, respectively). According to the AFCA (Ageing Fly Cell Atlas; <https://hongjielilab.org/afca/>) portal from the Hongjie Li laboratory, the “escort cell” group in the single cell database expresses *ptc* but very few cells express *sns* or *hbs*. Similarly, markers that should be expressed in all escort cells such as *Wnt4* (a TFC-CpC-EC marker) or *fax* (Mottier-Pave et al. *Dev. Biol.*, 2016; Decotto + Spradling, *Dev. Cell*, 2005; Rust et al., *Nat. Comms.*, 2020) are expressed only in a subset of the cells belonging to the “escort cell” cluster in the single cell Ageing Fly tSNE maps (see below). In addition, there seems to be some overlap in the “pre-follicle cells/follicle stem cells” cluster and that of “escort cells”. Thus, perhaps what is finally considered the “escort cells” cluster is larger and contains more than just the real ECs identified in their study? Perhaps the single-cell data may require further sub-clustering? Please find below a number of screenshots related to the abovementioned results.

Cell Type:

follicle stem cell and prefollicle cell

Dimension Reduction:

UMAP t-SNE

Toggle graphics controls

t-SNE by Age

Marker Gene

Aging Feature

DE Analysis

Age: 5

Age: 30

Age: 50

Age: 70

Gene name:

Wnt4

Cell Type:

escort cell

Dimension Reduction:

UMAP t-SNE

Toggle graphics controls

Toggle Column Filters

Reset Filter

Show 10 entries

Search:

Gene name	P-value	FDR	Log2FC	Cell Type

5- Finally, it may be that the results are technology-dependent. Both experimental setups are not completely comparable, but we were assuming a proportion of genes would be shared among both datasets. The fact that this is not the case could indicate that the context of sample isolation, treatment and analysis is critical for the final data.

Since this is a negative result and without a clear explanation, we prefer not to include this discussion in the text but we are open to suggestions from the reviewers.

Detailed response to the comments from reviewer 1

Reviewer 1: In this revised version of the manuscript, the authors have addressed all our comments. While the manuscript itself has not undergone significant changes in terms of data or conclusions, several important points have been improved. The authors have thoroughly addressed our criticisms in their rebuttal letter and have included multiple significant new experiments in this revised version.

Our primary criticism was that the manuscript initially seemed somewhat misleading. The authors described changes in gene expression and alternative splicing during the aging of specific cell populations (via RNA-seq) but then conducted mechanistic experiments (loss-of-function or gain-of-function) that did not correspond to the dynamics observed in their RNA-seq data. The authors have now clarified that they are not attempting to replicate aging and explicitly stated that their experiments "did not recapitulate aging." This critical point has been made much clearer.

We still have a few minor remaining points:

R1.1: Line 236 : « First, the amounts of Smu1 mRNA in ECs were 100-400-fold higher than in TFCs-CpCs. » This is not shown.

The reviewer is correct. We have modified the text to say 40-150 (the values in the Y axes of Figs. 4E and F are different. Values range from 2,5-3 copies/ng of cDNA in TFCs-CpCs to 100-400 in ECs).

We understand that the authors have already changed the fold changes, but it still does not include the data at 3weeks in TFC/CpC, which is closed to zero (Figure 4d). Why is this data point not taking into account?

The reviewer is right. We have changed the text to reflect the fact that Smu1 levels in TFC/CpC range from 0 to 3 and in ECs from 100 to 400.

R1.2: The genetic background is not discussed: Cell adhesion, cell polarity and cytoskeleton GE genes are a large part of altered genes in GE and AS experiment in TFs and CpCs populations. Does overexpression of tau::GFP misregulate such genes? For example, RT-qPCR of appropriate genes in *bab1>tau::GFP* and control ovaries could clarify this point..

*This is an important point that we discussed thoroughly while defining the strategy of our approach and that we addressed experimentally. We believe we did the right controls. First, flies expressing tau::GFP with any of the Gal4 drivers utilized were perfectly viable, healthy and fertile and possessed normal-looking ovaries. In fact, we keep stocks containing both the driver and the UAS-tau::GFP construct. Second, we scored GSC numbers and Fas2 localisation in CpC-CpC contacts of *bab1-Gal4*, *hh-Gal4*, *bab1-Gal4 UAS-tau::GFP* and *hh-Gal4 UAS-tau::GFP* 4-week-old flies and could not detect significant differences between them. Thus, tau::GFP expression in niche cells of 4 week-old females does not affect niche activity, at least as measured by GSC numbers and 1D4 levels. Third, our study identified differentially expressed genes in old versus young niche cells but that were treated in the same conditions, i.e. with the same cell sorting*

protocol and with the same genetic background (expressing tau::GFP). We agree with the reviewer that, should we be studying cells expressing tau::GFP and comparing them with tau::GFP-non expressing cells, we should perform additional controls. In our case, however, the genetic background in both conditions is the same and the expression of the GFP fusion seems to have no deleterious effect on niche activity.

This is a convincing and interesting answer, which should be included in the manuscript. It could be included in the Material and Methods section.

In accordance with the reviewer's suggestion, we have added the following paragraph to the M+M section: "We performed a series of tests to make sure the expression of tau::mGFP6 did not affect niche activity. First, flies expressing tau::GFP with any of the Gal4 drivers utilized were perfectly viable, healthy and fertile and possessed normal-looking ovaries. In fact, flies containing both the driver and the UASp-tau::GFP construct were kept as stocks. Second, we scored GSC numbers and Fas2 localisation in CpC-CpC contacts of bab1-Gal4, hh-Gal4, bab1>tau::GFP and hh>tau::GFP 4-week-old flies and could not detect significant differences between them. Thus, tau::GFP expression in 4 week-old niche cells does not impact their function, at least as measured by GSC numbers and 1D4 levels."

R1.3: We have added this information to the text and have modified Figs. 7 (panel H) and S7 (panel E).

We could not find panel E on Figure S7.

I have double checked that panel S7E and its correspondent fig. legend were included in the 2nd submission. Thus, this is correct in this final submission too.

R1.4: Several genotypes are missing or wrong for examples:

1) sup Figure S5D : genotypes are missing on line 644 in experimental genotypes

Well spotted! This has been corrected. Thank you.

2) sup Figure S9: wrong genotypes, I guess it is hh>Fas2i and not hh>Smu1i, as indicated in the supplementary figure 9 legends.

No, they are not wrong. I have confirmed that genotypes and fig. legend are correct. Panels A-D in Fig. S9 correspond to Smu1 RNAi experiments and panel E to Fas2 overexpression.

Etc...

I have gone through all of the genotypes and fig. legends. Apart from the lack of Fig. S5D genotypes (already corrected; see point R1.4), the rest of the figs. were correct.